# STEADYTHOUGHT:
# MITIGATING LLM UNDER-THINKING VIA THOUGHT-LEVEL PREFERENCE OPTIMIZATION

## ABSTRACT

Flexible switching between reasoning trajectories (i.e., thoughts switching) has significantly enhanced the reasoning capabilities of Large Reasoning Models (LRMs). However, existing models often switch excessively yet fail to sustain promising reasoning thoughts——a phenomenon termed "under-thinking". While recent efforts suppress switching to mitigate this, such over-correction may discard valuable trajectories. To address this challenge, we propose **Steady Thought** (**ST**), a novel thought-level preference optimization framework. ST first segments model responses into thought sequences then guides the model to complete reasoning from these thoughts without further switching, generating coherent trajectories. Finally, ST performs thought-level preference optimization by treating the newly generated response as preferred and the original one as dis-preferred. Experiments across multiple models and datasets show that ST effectively mitigates under-thinking. It reduces output length by up to 39.3% while improving accuracy by up to 5.3%, with strong generalization. Further analysis confirms that ST leads to more rational switching and deeper exploration of solution thoughts.

## 1 INTRODUCTION

Nowadays, Large Reasoning Models (LRMs), such as DeepSeek-R1 (DeepSeek-AI et al., 2025) and GPT-o1 (OpenAI, 2024), have demonstrated strong reasoning capabilities across complex tasks. Their success stems from human-like slow thinking and reflective behaviors, which enable flexible switching between reasoning strategies—a process termed *thought switching* (Zeng et al., 2025; Muennighoff et al., 2025; Yang et al., 2025; Chen et al., 2025b). This adaptability fosters exploration of diverse reasoning paths, yielding more accurate and robust inference.

However, recent studies have shown that LRMs tend to switch thoughts too frequently (Wang et al., 2025a; Ding et al., 2025; Chen et al., 2025a) and often fail to follow the promising thoughts—a phenomenon termed "under-thinking" (Wang et al., 2025c). Specifically, Figures 1a and 1b illustrate the initial emergence of correct thoughts during the models' reasoning processes. They reveals that the models often find a reliable thought early (which can lead to the correct answer), but still proceed with numerous additional thought switches. This issue roots from the lack of ability to recognize and commit to promising reasoning trajectories, which ultimately hinders the depth, coherence, and overall quality of reasoning. Furthermore, such excessive switching leads to substantial inefficiencies, resulting in unproductive exploration and a waste of computational resources.

To mitigate the shallow reasoning caused by frequent thought switching, existing approaches primarily aim to suppress the switching behavior during inference. For example, some methods (Wang et al., 2025c;a; Ding et al., 2025) assume that the thought switching often starts with certain special tokens (e.g., "alternatively" or "wait") and suppressing under-thinking by lowering the probabilities of these tokens during decoding or reducing their rewards in training. On the other hand, Chen et al. (2025a) takes a different approach by operating in the representation space: it steers the model's hidden states away from a learned "switching" vector and toward a "reasoning" vector, thereby suppressing switching behavior in a more structured manner. While effective in reducing excessive switching, these methods apply suppression globally, potentially limiting the model's flexibility to explore alternative reasoning thoughts when necessary. These limitations highlight the need for a

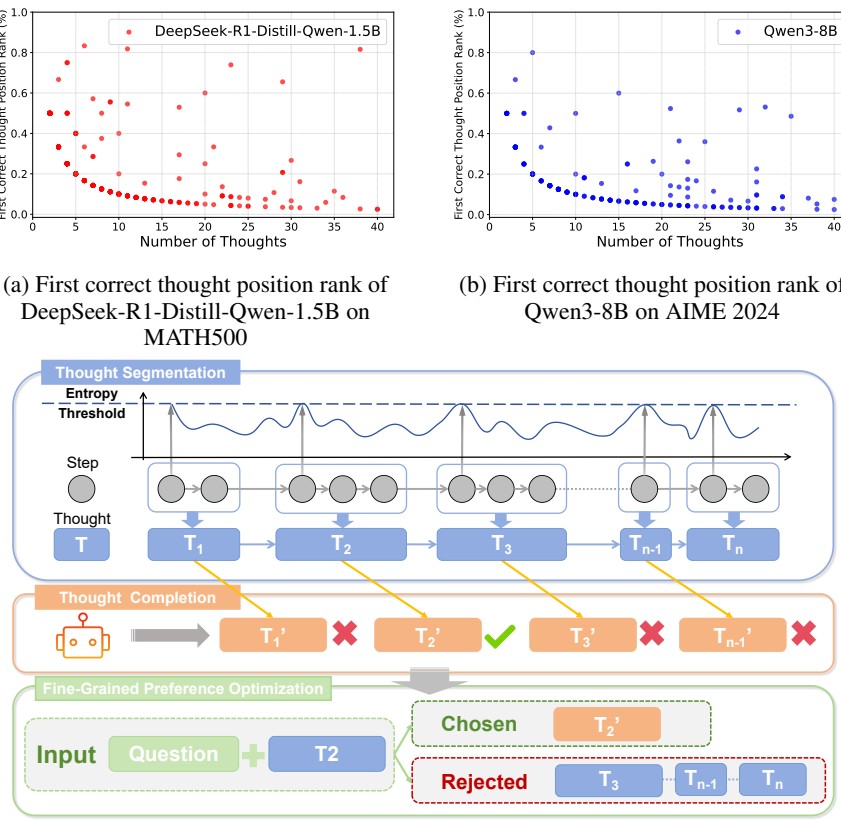

(a) First correct thought position rank of DeepSeek-R1-Distill-Qwen-1.5B on MATH500

(b) First correct thought position rank of Qwen3-8B on AIME 2024

(c) Overview of Steady Thought framework

Figure 1: (a) Rank of the first correct thought position for DeepSeek-R1-Distill-Qwen-1.5B on MATH500. (b) Rank of the first correct thought position for Qwen3-8B on AIME2024. (a) and (b) plot the percentile rank of the first correct thought in a thought sequence against the total number of thoughts segmented from each model response. (c) Overview of the ST framework, which operates in three core stages: **(1) Thought Segmentation:** Segmenting the thinking part of the response at a thought level based on changes in entropy. **(2) Thought Completion:** Guiding the model to continue writing for each thought without any thoughts switching by reducing the logits of words such as "wait" and "alternatively." **(3) Fine-Grained Preference Optimization:** Constructing preference optimization data pairs based on the correctness of the completion, and optimizing the model using the STPO algorithm.

more selective mechanism—one that preserves the ability to explore new reasoning thoughts when the current trajectory is unpromising, while encouraging deeper commitment to a thought when it shows promise.

To this end, we propose **Steady Thought** (**ST**), a novel thought-level preference optimization framework. ST aims to guide the model to consistently pursue high-potential reasoning thoughts while preserving its ability to explore necessary alternatives. As show in Figure 1c, ST operates in three stages: (1) **Thought Segmentation:** the model's thinking part wrapped in <think> tags of response is segmented into a sequence of thoughts by integrating step-level split and entropy-based thought switching detection. (2) **Thought Completion:** The target model is guided through logits control to generate new reasoning content based on a specific thought, thereby progressing toward the final answer. This process keeps the model on track and allows for the in-depth development of promising thoughts. (3) **Fine-Grained Preference Optimization:** By treating the newly generated content that leads to the correct answer as chosen, and the original as rejected, we perform a thought-level preference optimization named **Steady Thought Preference Optimization** (STPO) to encourages the model to favor reasoning thoughts that demonstrate more consistent progress to-

ward the correct final answer, but at the same time, without detriment to the model's capability for preliminary exploration of various possible reasoning thoughts.

Our main contributions are threefold:

- We analyzed the specific manifestations of the under-thinking phenomenon and formalized it as a preference optimization problem.
- We propose **Steady Thought** (ST), a novel thought-level preference optimization framework that encourages models to develop and stick to high-potential reasoning thoughts without compromising their flexibility to explore alternative reasoning trajectories.
- We validate the effectiveness of ST through extensive experiments across multiple models and reasoning benchmarks, with results demonstrating effective accuracy improvements of up to 5.3% and significant token reductions ranging from 19.0% to 39.3%.

## 2 BACKGROUND

### 2.1 PROBLEM FORMULATION

We formalize the reasoning process of a Large Reasoning Model (LRM) by modeling its response $\mathbf{y}$ to a question $\mathbf{x}$ as a trajectory of distinct thoughts:

$$\mathbf{y} = (T_1, T_2, \ldots, T_n), \tag{1}$$

where each thought $T_i$ represents a coherent segment of reasoning. After generating a prefix of $i$ thoughts, denoted $\mathbf{P}_i = (\mathbf{x}, T_1, \ldots, T_i)$, the policy $\pi_\theta$ defines a conditional probability distribution over all possible subsequent trajectories.

The problem of **under-thinking** arises at a critical decision point. After the model generates a **promising thought** $T_i$ (one that *can* lead to a correct answer), it faces a choice. Within the set of possible next trajectories, we are interested in two specific types:

- The **Commit Trajectory** ($\tau_{\mathbf{c}}$): The ideal trajectory where the model commits to and correctly completes the promising thought $T_i$. We denote this completed sequence as $T_i'$.
- The **Switch Trajectory** ($\tau_{\mathbf{s}}$): The suboptimal trajectory observed in the data, where the model abandons $T_i$ and switches to a new, often wasteful line of reasoning $(T_{i+1}, \ldots, T_n)$.

Our objective is to align the model's policy $\pi_\theta$ with the preference for the commit trajectory over the switch trajectory, i.e., $\tau_{\mathbf{c}} \succ \tau_{\mathbf{s}}$. To formalize this, we define a latent **Steadiness Score** $S_\pi(\tau|\mathbf{P}_i)$, which quantifies the policy's inclination towards a specific trajectory $\tau$ given the prefix $\mathbf{P}_i$. The preference $\tau_{\mathbf{c}} \succ \tau_{\mathbf{s}}$ implies that the score of the commit trajectory should be higher.

Following the Bradley-Terry model, the probability of this preference can be expressed through the difference in scores:

$$P(\tau_{\mathbf{c}} \succ \tau_{\mathbf{s}}|\mathbf{P}_i) = \sigma(S_\pi(\tau_{\mathbf{c}}|\mathbf{P}_i) - S_\pi(\tau_{\mathbf{s}}|\mathbf{P}_i)) \tag{2}$$

where $\sigma$ is the sigmoid function. This abstract scoring function provides a powerful new lens through which to view the problem. Crucially, it can be directly instantiated by the LRM's own log-probabilities. Preference optimization methods like Direct Preference Optimization (DPO) (Rafailov et al., 2024) and Simple Preference Optimization (SimPO) (Meng et al., 2024) allow us to directly optimize the policy $\pi_\theta$ such that its implicit scoring function, where $S_\pi(\tau|\mathbf{P}_i) := \log \pi_\theta(\tau|\mathbf{P}_i)$, satisfies the desired preference relationship. This provides a principled way to train the model to be a more steadfast and effective reasoner.

### 2.2 PREFERENCE OPTIMIZATION METHODS

DPO significantly advanced the alignment of language models with human preferences by leveraging the Bradley-Terry preference model to bypass the complex reward modeling and policy optimization of traditional reinforcement learning. However, DPO is known to be sensitive to length bias, as its objective can implicitly favor longer sequences that naturally attain higher log-likelihoods. SimPO addresses this limitation by introducing a length-normalized reward with a

target margin, effectively mitigating length bias without requiring a reference model. Given the preference data $\mathcal{D} = \{(\mathbf{x}, \mathbf{y}_w, \mathbf{y}_l)\}$, where $\mathbf{x}$ is the input prompt, $\mathbf{y}_w$ is the preferred response, and $\mathbf{y}_l$ is the dispreferred response. SimPO optimizes the following objective:

$$\mathcal{L}_{\text{SimPO}}(\pi_\theta) = -\mathbb{E}_{(x, y_w, y_l) \sim \mathcal{D}} \left[ \log \sigma \left( \frac{\beta}{|y_w|} \log \pi_\theta(y_w|x) - \frac{\beta}{|y_l|} \log \pi_\theta(y_l|x) - \gamma \right) \right]. \quad (3)$$

where $\sigma$ is the sigmoid function, with $\beta$ as a temperature parameter controlling sensitivity to preference differences and $\gamma$ as the target reward margin for the model to achieve.

## 3 METHOD

We propose Steady Thought (ST), a preference optimization framework that enables models to learn to stick to promising reasoning thoughts. As illustrated in Figure 1c, ST consists of three stages: 1) Thought Segmentation, 2) Thought Completion, and 3) Fine-Grained Preference Optimization, to teach models when to switch thoughts and when to persist.

### 3.1 THOUGHT SEGMENTATION

To segment the model's response into thought-level units, we employ entropy as a quantitative metric for assessing the model's confidence. The underlying principle is that a high entropy value indicates uncertainty in the model's predictive distribution over the next token, which typically occurs when it re-plans or explores a new reasoning trajectory. Conversely, low entropy signifies high confidence in the current output. Some recent studies indicate that tokens with high entropy often play a decisive role in determining reasoning trajectories (Wang et al., 2025b). Therefore, a sudden spike in entropy can serve as an effective signal for a thought switch. The entropy is calculated as follows:

$$H(P) = -\sum_x P(x) \log P(x) \quad (4)$$

where x is a token in the sequence, and P(x) represents the model's predicted probability for that token.

Operationally, we first pre-segment the response into candidate steps using the common logical delimiter ".\n\n". We then compute the entropy for each token. A thought switch is identified if any of the initial tokens at the beginning of a candidate step exhibits entropy exceeding a predefined threshold, marking the start of a new thought. All subsequent steps are merged into this thought until the next threshold-exceeding step is encountered. This combines initial segmentation with entropy-based detection to partition each response $\mathbf{y}$ into a sequence of thoughts:

$$\mathbf{y} = (T_1, T_2, \ldots, T_n) \quad (5)$$

The granularity of this segmentation is controlled by the entropy threshold. We determined the optimal threshold through hyperparameter tuning to balance the detection of meaningful switches against over-segmentation. An excessively high threshold may miss subtle reasoning adjustments, while an overly low one could fragment coherent reasoning. For detailed experiments, see Section 4.4.3.

### 3.2 THOUGHT COMPLETION

The purpose of the second stage is to acquire a correct, self-generated completion of the thought previously segmented without any switches. We first predefine specific trigger words (e.g., "wait" and "alternatively") that signal a thought switch. During decoding, we then sharply decrease the logits for these words, effectively suppressing their selection by driving their prediction probability close to zero. For each thought $T_i$ segmented in the previous stage, we apply the method to the model to continue solving the problem in conjunction with the corresponding question $Q$, yielding the completion:

$$T_i' = \text{Model}(Q, T_i), \quad (6)$$

where Model is target model and $T_i$ contains no thought switches and outputs a final answer. By evaluating the correctness of that final answer, we can determine whether the thought was a valid one. We discuss the consumption generated by this stage in the Appendix E

### 3.3 FINE-GRAINED PREFERENCE OPTIMIZATION

A significant limitation of holistic preference optimization is that it treats entire reasoning chains as monolithic blocks. For complex problems, an incorrect response often contains a substantial sequence of correct initial reasoning. Rejecting such a response in its entirety discards the valuable correct portion and provides a noisy, unfocused learning signal to the model.

To address this, our approach provides more granular supervision by focusing on the critical juncture where the reasoning diverges. We aim to teach the model to commit to a promising thought once it has been identified, rather than abandoning it. Specifically, an answer $\mathbf{y}$ can be decomposed into a sequence of thoughts $\mathbf{y} = (T_1, \ldots, T_n)$. When we identify a promising thought $T_i$, the optimization then hinges on the trajectory the model takes from this point forward, conditioned on both the original question $Q$ and the thought $T_i$ itself.

The preference pair is constructed based on the continuations from this shared context:

$$\text{Chosen Response}: \quad \mathbf{y}_w = T_i' \qquad \text{(The full, correct completion)}$$

$$\text{Rejected Response}: \quad \mathbf{y}_l = (T_{i+1}, T_{i+2}, \ldots, T_n) \qquad \text{(The subsequent wasteful thoughts)}$$

Our Steady Thought Preference Optimization (STPO) objective is to train the model to prefer the chosen response over the rejected one, conditioned on this specific, fine-grained context. Inspired by the reference-free and length-normalized objective of SimPO, the STPO loss is formulated as:

$$\mathcal{L}_{\text{STPO}}(\pi_\theta) = -\mathbb{E}_{(Q, T_i, \mathbf{y}_w, \mathbf{y}_l) \sim \mathcal{D}} \left[ \log \sigma \left( \frac{\beta}{|\mathbf{y}_w|} \log \pi_\theta(\mathbf{y}_w | Q, T_i) - \frac{\beta}{|\mathbf{y}_l|} \log \pi_\theta(\mathbf{y}_l | Q, T_i) - \gamma \right) \right] \tag{7}$$

This formulation provides a crucial distinction from conventional preference tuning. The learning signal, embodied by the conditional log-probabilities $\log \pi_\theta(\cdot | Q, T_i)$, is applied directly at the point of divergence. It forces the model to learn not just what a good final answer looks like, but to recognize and commit to a promising intermediate thought. This targeted, thought-level supervision is key to effectively mitigating the model's tendency to "under-think" and abandon viable reasoning paths.

## 4 EXPERIMENTS

### 4.1 DATASETS

We selected the omni-math (Gao et al., 2024) dataset as the source of our training data. This dataset contains thousands of problems at the level of the International Mathematical Olympiad, which are categorized by difficulty. We sampled problems from various difficulty levels and used a target model (e.g., Qwen3-8B) to perform inference and generate responses.

We evaluated our method on four datasets: (1) MATH-500 (Hendrycks et al., 2021): A math test set containing 500 problems with difficulty levels ranging from 1 to 5, covering multiple mathematical domains such as algebra, geometry, and number theory. (2) AIME 2024 (MAA Committees, 2024): A high-difficulty math competition dataset based on the American Invitational Mathematics Examination (AIME), consisting of 30 problems covering mathematical branches such as Algebra, Geometry, Number Theory, and Combinatorics. (3) GSM8K (Cobbe et al., 2021): A benchmark dataset featuring high-quality, linguistically diverse math word problems. It is used to evaluate the multi-step reasoning capability of language models. While based on simple arithmetic and basic algebra, solving these problems requires 2 to 8 sequential steps. The standardized test set consists of 1,319 problems. (4) LiveCode (Jain et al., 2024):A dataset for evaluating the code capabilities of LLMs, containing 400 problems collected from competitive programming websites like LeetCode, AtCoder, and Codeforces. We selected this dataset as an out-of-distribution (OOD) dataset to test the model's generalization ability after training.

### 4.2 BASELINE AND METRICS

We compare Steady Thought against three base models (DeepSeek-R1-Distill-Qwen-1.5B (DeepSeek-AI et al., 2025), Qwen3-8B (Qwen Team, 2024) and DeepSeek-R1-Distill-Qwen-14B (DeepSeek-AI et al., 2025)) as well as three test-time efficiency methods, namely NoThink, NOWAIT and SEAL.

- NoThink: Ma et al. (2025) make the model skip the thinking process and output the response directly by adding the `<think>` tag after the `</think>` tag in the prompt.
- NOWAIT: Wang et al. (2025a) reduces the logits values of certain keywords that represent reflection (e.g., "wait") during decoding, making them almost impossible to output.
- SEAL: Chen et al. (2025a) first collects multiple responses and categorizes them into three types of thoughts: execution, reflection, and transition. A differential vector S is derived in the latent space between execution vectors and the combination of reflection/transition vectors. During decoding, the hidden states are modified as $\tilde{H} = H + \alpha \cdot S$, biasing the hidden layer toward execution-oriented thoughts.

We report the accuracy and average token count performance for each task. We took the average of eight test runs for the AIME 2024 test set and two runs for the LiveCode test set.

## 4.3 MAIN RESULTS

As shown in Table 1, the Steady Thought method performed well on three reasoning models of different architectures and sizes. It effectively reduced the number of tokens generated by the models while maintaining or even improving their accuracy. For example, on the DeepSeek-R1-Distill-Qwen-1.5B model, ST increased the average accuracy across four datasets by 1.9% while reducing the average token count by 24.9% compared to the base model. Similarly, on the Qwen3-8B model, it boosted the average accuracy by 3.12% and reduced the average token count by 23.6%. ST achieved a successful performance improvement of 2.52% on average for the DeepSeek-R1-Distill-Qwen-14B model, while simultaneously reducing the average output length by 17.3%.

It is noteworthy that the LiveCode dataset serves as an OOD test set, as our models were trained exclusively on a mathematical training dataset. Despite this, ST still achieved positive results on LiveCode. For instance, it improved the Qwen-8B's accuracy by 5.3% and reduced its token count by 19.0%. Furthermore, it achieved similar gains on a larger 14B model, boosting its accuracy by 4.2% and reducing its output length by 14.2%. This suggests that the ST effectively teaches the model a more precise pattern of thought switching and retention, rather than simply memorizing the data, thereby improving its generalization ability. We have provided specific examples of the trajectories in Appendix B to demonstrate the effectiveness of ST.

Table 1: Experimental results on two large reasoning models.

| Method | MATH-500 | | AIME 2024 | | GSM8K | | LiveCode | | Overall | |
|---|---|---|---|---|---|---|---|---|---|---|
| | Acc(%)↑ | Tokens↓ | Acc(%)↑ | Tokens↓ | Acc(%)↑ | Tokens↓ | Acc(%)↑ | Tokens↓ | Acc(%)↑ | Tokens↓ |
| *DeepSeek-R1-Distill-Qwen-1.5B* | | | | | | | | | | |
| *Vanilla* | 82.0 | 4385 | 27.5 | 11273 | 81.9 | 1448 | 30.3 | 9623 | 55.43 | 6682 |
| *NoThink* | 65.8 | 749 | 8.7 | 3185 | 53.6 | 263 | 20.7 | 813 | 37.20 (-18.23) | 1252 (-81.3%) |
| *NOWAIT* | 80.6 | 2433 | 20.8 | 7000 | 66.1 | 2078 | 28.3 | 4927 | 48.95 (-6.48) | 4109 (-38.5%) |
| *SEAL* | 82.6 | 3252 | 25.4 | 9120 | 79.7 | 860 | 29.5 | 7948 | 54.30 (-1.13) | 5295 (-20.8%) |
| *Steady Thought* | 84.4 | 2809 | 31.2 | 8606 | 81.3 | 1254 | 32.4 | 7398 | 57.33 (+1.9) | 5016 (-24.9 %) |
| *Qwen3-8B* | | | | | | | | | | |
| *Vanilla* | 91.4 | 4724 | 62.1 | 10895 | 95.6 | 1759 | 71.8 | 7112 | 80.23 | 6122 |
| *NoThink* | 85.2 | 933 | 25.8 | 3504 | 93.6 | 289 | 45.6 | 584 | 62.55 (-17.68) | 1327 (-78.3%) |
| *NOWAIT* | 61.0 | 13274 | 26.3 | 14333 | 73.3 | 12369 | 75.5 | 5226 | 59.03 (-21.20) | 11300 (+84.6%) |
| *SEAL* | 92.2 | 4034 | 58.8 | 10372 | 95.9 | 1421 | 83.4 | 6414 | 82.58 (+2.35) | 6940 (-8.4%) |
| *Steady Thought* | 94.4 | 2869 | 65.8 | 8742 | 96.1 | 862 | 77.1 | 5759 | 83.35 (+3.12) | 4558 (-25.5 %) |
| *DeepSeek-R1-Distill-Qwen-14 B* | | | | | | | | | | |
| *Vanilla* | 93.6 | 3349 | 60.4 | 8974 | 94.8 | 894 | 70.1 | 6789 | 79.73 | 5001 |
| *NoThink* | 41.7 | 824 | 27.1 | 3279 | 90.1 | 256 | 44.0 | 708 | 50.73 (-29.00) | 1266 (-74.7%) |
| *NOWAIT* | 75.6 | 3314 | 33.8 | 9431 | 86.3 | 936 | 64.3 | 5099 | 65.00 (-14.73) | 4695 (-6.1%) |
| *SEAL* | 92.6 | 3253 | 60.8 | 8831 | 94.7 | 880 | 75.1 | 6706 | 80.80 (+1.07) | 4917 (-1.7%) |
| *Steady Thought* | 94.2 | 2455 | 65.4 | 7554 | 95.1 | 715 | 74.3 | 5825 | 82.25 (+2.52) | 4137 (-17.3%) |

## 4.4 IN-DEPTH ANALYSIS AND ABLATION STUDIES

### 4.4.1 ANALYSIS OF IN-DEPTH EXPLORATION ABILITY

ST improves the model's capacity for in-depth exploration of promising thoughts. This effect is evident in two key metrics: First, the model's output becomes more concise, with a reduction in average output length. Second, the final thought——the one leading to the definitive answer——constitutes a significantly larger proportion of the overall response. This suggests that once the model identifies

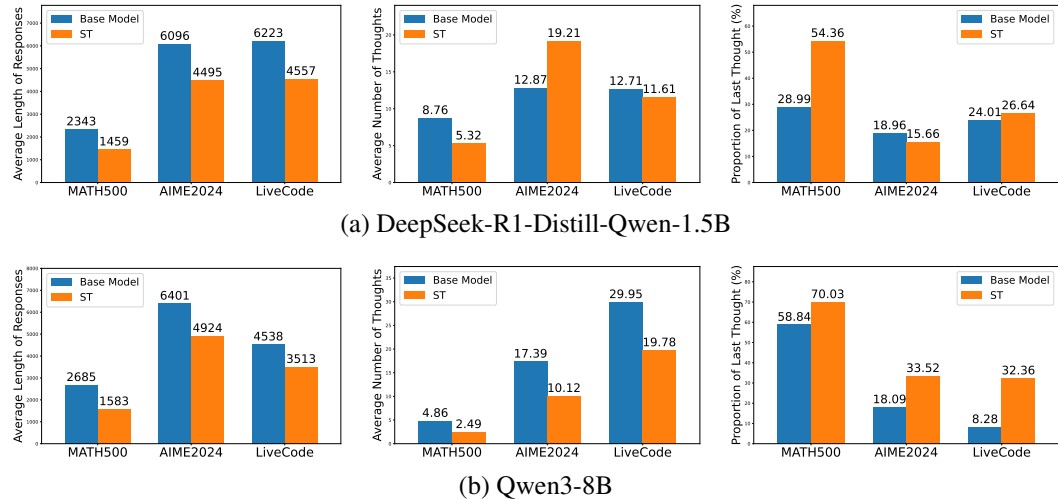

Figure 2: Comparison of model behavior across three metrics: average response length (left), average number of thoughts (middle), and proportion of last thought (right). The first row shows results for DeepSeek-R1-Distill-Qwen-1.5B, while the second row shows Qwen3-8B. Blue bars represent the base model, orange bars ST method.

a promising thought with the potential to lead to the correct solution, it is better equipped to pursue that thought thoroughly instead of frequently switching its focus.

By applying the methodology described in Section 3.1, we partitioned and quantified the thought processes from the model's responses both before and after ST. As Figure 2 illustrates, ST consistently produced shorter average outputs across the all of datasets. Furthermore, in most cases, the average number of thoughts generated by the model also decreases correspondingly. However, when smaller models tackle high-difficulty problems, they tend to increase the frequency of thought transitions to find the optimal solution. For example, when addressing the challenging AIME 2024 dataset, DeepSeek-R1-Distill-Qwen-1.5B generated a greater number of thoughts under the ST method compared to the base model. This increase led to improved accuracy and shorter overall response length. For less challenging problems, both the 1.5B and 8B models tended to produce fewer thoughts. Additionally, the final thought consistently accounted for a larger proportion of the total response. These experiments provide compelling evidence that the ST method significantly enhances the model's in-depth exploration capability.

### 4.4.2 ANALYSIS OF THINKING SWITCHING ABILITY

ST enhances the model's accuracy in determining when to switch its thoughts of reasoning, making its thought transitions more purposeful. This improvement is primarily evidenced by two observations: the model significantly reduces its total output length while maintaining high performance, and the proportion of correct intermediate thoughts decreases before the final answer is reached. In a thought chain, any correct intermediate thought that is subsequently abandoned and switched to another path constitutes an Invalid Switch. Thus, the number of correct intermediate thoughts is equal to the number of Invalid Switches. Consequently, we aim to demonstrate the reduction in the phenomenon of ineffective switching by showing the decrease in the proportion of correct thoughts after the model undergoes ST training.

As shown in Table 2, we used the methods from Section 3.1 and 3.2 to calculate the proportion of correct thoughts preceding the final one, both before and after ST. On both models and across both datasets, the ST-trained model consistently achieved a lower rate of correct intermediate thoughts. This precisely demonstrates that the model's decision-making has become more precise—it reduces instances of unnecessary thought-switching, allowing for a more reasoned and thorough preliminary exploration. We further discuss the benefits that ST brings to the model through thought-level preference optimization in the Appendix C.

Table 2: Percentage of correct thoughts (PCT) generated by the model before and after applying ST.

| Method | MATH500 PCT(%)↓ | AIME 2024 PCT(%)↓ |
|---|---|---|
| DeepSeek-R1-Distill-Qwen-1.5B | 54.90 | 14.50 |
| + Steady Thought | **40.40** | **7.90** |
| Qwen3-8B | 73.18 | 45.20 |
| + Steady Thought | **67.74** | **39.00** |

### 4.4.3 ANALYSIS OF DIFFERENT ENTROPY THRESHOLD

As mentioned in Section 3.1, the entropy threshold setting directly influences the granularity of thought segmentation, which in turn affects the model's learning. An excessively high threshold leads to coarse-grained segmentation, potentially missing critical thought steps which are able to reach the correct answer. This also reduces the amount of data available for the model to learn from. Conversely, a threshold that is too low can split apart complete thoughts, making it difficult to identify a correct reasoning thought that should have been whole. We analyzed the segmentation results of the DeepSeek-R1-Distill-Qwen-1.5B based on different entropy threshold settings, as detailed in Table 3.

From the table, we can see a trade-off in thought segmentation. A lower threshold, which leads to finer-grained segmentation, results in a drop in the proportion of correct thoughts. Conversely, a higher threshold creates coarser segments. While this improves the proportion of correct thoughts, it reduces the amount of available data for training. Ultimately, we found a threshold of 3.0 to be optimal for the DeepSeek-R1-Distill-Qwen-1.5B model which was used in our experiments, as it yielded the best overall performance. We provide threshold tuning results on more models and datasets in the appendix D.

Table 3: Comparison of thought segmentation results under varying thresholds. NT: number of segmented thoughts. PCT: proportion of correct thoughts.

| Base model | Threshold | ST Metric | | MATH500 | | AIME 2024 | |
|---|---|---|---|---|---|---|---|
| | | NT | PCT | Acc(%)↑ | Tokens↓ | Acc(%)↑ | Tokens↓ |
| | 2.8 | 20650 | 21.93% | 83.4 | 3854 | 29.2 | 9252 |
| DeepSeek-R1-Distill-Qwen-1.5B | **3.0** | 10444 | 24.69% | **84.4** | **2809** | **31.2** | **8606** |
| | 3.2 | 4355 | 27.99% | 83.6 | 3657 | 28.3 | 10516 |

### 4.4.4 ANALYSIS OF DIFFERENT TRAINING METHOD

Table 4: Comparison of different training method results.

| Training Method | MATH500 | | AIME 2024 | |
|---|---|---|---|---|
| | Acc(%)↑ | Tokens↓ | Acc(%)↑ | Tokens↓ |
| DeepSeek-R1-Distill-Qwen-1.5B | 82.2 | 4385 | 27.5 | 11273 |
| SFT | 80.4 | 2650 | 22.9 | 7169 |
| DPO | 82.6 | 4273 | 30.8 | 10701 |
| STPO | 84.4 | 2809 | 31.2 | 8608 |

We analyzed the impact of different training methods in ST's last stage (or phase) on model performance, and the results are shown in Table 4. In the fine-tuning approach, we used the x from the preference data pairs as the input and the chosen response as the output. This method tends to make the model memorize specific data rather than learn the underlying reasoning patterns we want it to acquire. Consequently, although it has learned the characteristics of "chosen": very short content, its performance on highly difficult or OOD data is subpar.

In contrast, we want the model to accomplish two things: to deeply explore promising solution thoughts while also effectively rejecting useless switching in its thought process. This dual-focus optimization capability is unique to preference optimization methods and cannot be achieved through simple fine-tuning alone. When constructing our preference data, the complete response that follows a specific "thought" is typically designated as the rejected part, while the completion of that same thought is the chosen part. This often results in the rejected portion being significantly longer than the chosen portion. Although Direct Preference Optimization (DPO) (Rafailov et al., 2024) is widely popular for its simplicity and efficiency, its training effectiveness is compromised by its sensitivity to the stark length differences between chosen and rejected responses. SimPO introduces "length-normalized rewards," which effectively eliminates the impact of these length differences, allowing the model to better learn the deep patterns embedded within the data. Drawing inspiration from SimPO, we contribute STPO, a novel preference optimization framework operating at the level of thoughts. As a result, compared to the other two methods, STPO not only effectively reduces the model's output length but also improves its performance.

# 5 RELATED WORK

## 5.1 OVER-THINKING AND UNDER-THINKING

Recent studies have shown that while Large Language Models possess strong reasoning capabilities, they often expend excessive unnecessary resources—a phenomenon known as over-thinkingQu et al. (2025); Sui et al. (2025). This primarily manifests in two forms: first, the model's responses contain redundant information; second, it struggles to allocate appropriate computational budgets to questions of varying difficulty, with the latter being particularly evident in O1-like models.To address the first issue, existing solutions include both non-training and training methods. Non-training methods involve carefully designed prompts to limit output lengthXu et al. (2025); Nayab et al. (2025) or adjusting the model's output during decoding to enhance reasoning concisenessQiu et al. (2024); Zhang et al. (2025b). Training methods improve the model's ability to provide concise answers by fine-tuning on brief chain-of-thought (CoT) data or incorporating additional rewards/penalties for output length during reinforcement learning (RL) trainingKang et al. (2024); Aggarwal & Welleck (2025).For the second issue, most approaches employ fine-tuning or RL to enable models to dynamically switch between fast-thinking and slow-thinking modes during reasoning based on question difficulty, thereby improving the balance between reasoning accuracy and efficiencyZhang et al. (2025a); Lou et al. (2025).Under-thinking represents another form of resource wastage, primarily caused by LRMs frequently making ineffective switching thoughts during reasoning, preventing the model from fully developing promising reasoning thoughtsWang et al. (2025c). Existing methods focus on directly suppressing the model's reasoning thoughts switching at either the token or representation level to mitigate this phenomenonDing et al. (2025); Chen et al. (2025a); Wang et al. (2025a); Ding et al. (2025). In contrast, our approach considers that some switching is necessary, and instead alleviates under-thinking by enhancing the model's ability to maintain promising reasoning thoughts.

## 5.2 PREFERENCE OPTIMIZATION

Preference optimization is a technique that adjusts AI model behavior through human feedback to make its outputs more accurate or better aligned with human values. For example, some Proximal Policy Optimization (PPO)-based methods optimize model performance by incorporating human feedback during the reinforcement learning phase to construct a reward model, thereby limiting the scope of policy updates Lightman et al. (2024); Luo et al. (2023). But the training process for PPO methods is highly complex and heavily reliant on the quality of human feedback data. Direct Preference Optimization (DPO) (Rafailov et al., 2024) method, which requires fewer training models and has a shorter process, is becoming increasingly popular. Many studies have extended DPO to different levels, such as the step level and token level, to optimize the model (Lu et al., 2024; Lai et al., 2024; Liu et al., 2024). However, DPO is not suitable for scenarios with significant length differences in preference pairs due to its sensitivity to data length. In contrast, Simple Preference Optimization (SimPO) (Meng et al., 2024) avoids the model from gaining rewards by exploiting length by introducing the average log probability as the optimization benchmark, thereby decoupling "quality" from "length". Inspired by SimPO, we have developed a finer-grained preference

optimization framework, enabling the model to learn thought-level responses to mitigate the under-thinking issue.

## 6 CONCLUSION

This paper presents Steady Thought (ST), a novel preference optimization framework designed to mitigate the under-thinking problem in Large Reasoning Models. To address the issue of models frequently abandoning promising reasoning thoughts, ST introduces a structured pipeline: Thought Segmentation, Thought Completion, and Fine-Grained Preference Optimization, which guides models to switch thoughts judiciously and explore promising thoughts deeply. Experimental results on diverse models and tasks show that ST successfully mitigates under-thinking by reducing unnecessary switches, leading to more focused reasoning while maintaining or even enhancing performance.

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

# A  IMPLEMENTATION DETAILS

## A.1  TRAINING AND INFERENCE DETAILS

For the 1.5B model, we trained with a learning rate of 5e-7, a batch size of 128, a $\gamma$ of 1.0, and a $\beta$ set to 2.0. For the 8B model, we used Low-Rank Adaptation (LoRA) and set the learning rate to 3e-5, while keeping other hyperparameters consistent with the 1.5B model. Both models had a context window of 6144 and were optimized using Adam. Our training framework incorporated DeepSpeed ZeRO-3 with CPU offload for memory optimization. We also implemented a warm-up ratio of 0.1 with a cosine learning rate scheduler. For all inference tasks, we maintained uniform generation parameters with $max\_tokens = 14336$, $temperature = 0.6$, and $top\_p = 0.95$.

## A.2  PREDEFINED TOKENS TO BE SUPPRESSED

The thought completion phase of the ST framework involves the application of a significant negative bias to the logits of tokens indicative of thought switches, thereby driving their occurrence probability to near zero. The predefined set of tokens targeted for this suppression is provided in Table 5.

Table 5: Predefined set of tokens targeted for suppression.

| | | | | | | | | |
|---|---|---|---|---|---|---|---|---|
| (any | -any | .Any | .any | :Any | :any | <Any | <any | Any |
| [Any | _any | any | ĠAny | (other | -other | .Other | .other | Other |
| _other | other | ĠOther | Ġother | Hmm | ĠHmm | Ġhmm | Alternative | alternative |
| ĠAlternative | Ġalternative | Another | another | ĠAnother | Ġanother | (check | -check | .Check |
| .check | /check | Check | _Check | _check | check | ĉCheck | ĉcheck | ĠCheck |
| Ġcheck | Alternatively | ĠAlternatively | Ġalternatively | "But | ,but | -but | .But | .but |
| But | _but | but | âĢ¦but | âĢĶbut | âĢĬBut | ĠBut | Ġbut | However |
| however | ĠHowever | Ġhowever | .Verify | .verify | Verify | _verify | verify | ĉverify |
| ĠVerify | Ġverify | Again | _again | again | ĠAgain | Ġagain | (wait | .Wait |
| .wait | /wait | Wait | _wait | wait | ĉwait | ĠWait | Ġwait | "Now |
| (now | -now | .Now | .now | Now | [now | _now | now | ĉnow |
| ĠNow | Ġnow | Ah | ah | aha | ĠAh | Ġah | "Oh | Oh |
| oh | âĢĬOh | ĠOh | Ġoh | .Maybe | Maybe | maybe | ĠMaybe | Ġmaybe |

# B  EXAMPLE OF TRAJECTORY

We demonstrate the stabilization effect brought by the ST method by showing the Qwen3-8B model's response trajectory on the same problem before and after its application as presented in Table 6 . We also illustrate the segmentation results based on entropy. Observing the trajectory reveals that, prior to applying ST, the model frequently switched its thought direction on this problem, ultimately failing to complete the final answer within the pre-set maximum token limit. However, after applying ST, the number of thoughts was reduced from 11 to 4, the token count decreased from 14,338 to 7,413, and the answer was correct.

# C  FURTHER ANALYSIS ON STEADY THOUGHT AND REASONING EXPLORATION

We provide a more detailed analysis discussing how ST improves the model's output efficiency and analyzing the change in the model's exploration capability after ST. Our analysis demonstrates that STPO primarily targets and reduces **invalid switching**, while preserving and even optimizing the ability for **necessary exploration**.

To clarify the mechanism of ST, we first define the two types of thought-switching behaviors we address:

- **Valid Switching (Reasonable Exploration, $F \rightarrow T$):** Switching from an **incorrect thought** ($F$) to the next thought, which represents the model undertaking necessary exploration to correct its trajectory. This behavior is valuable.

- **Invalid Switching (Unwarranted Abandonment, $T \rightarrow F$):** Switching from a **correct thought** ($T$) to the next thought, which represents the model failing to identify and persist

Table 6: An example of result based on thought segmentation.

**Problem**:
The polynomial $x^3 - 3x^2 + 4x - 1$ is a factor of $x^9 + px^6 + qx^3 + r$. Enter the ordered triple $(p, q, r)$.
**Reference Answer**: (6,31,-1)

BEFORE ST (14338 OUTPUT TOKENS:

Thought 1: Okay, so I need to find the ordered triple $(p, q, r)$ such that the polynomial $x^3 - 3x^2 + 4x - 1$ is a factor of $x^9 + px^6 + qx^3 + r$. ... Let me check if this makes sense.

Thought 2: Wait, let me verify this result by another method to be sure. Maybe by polynomial division or by considering that $x^9$ ... So perhaps my earlier approach is correct.

Thought 3: But let me check with the values I found. If $p = 6, \ldots$ Let me check if $x^3 - 3x^2 + 4x - 1$ divides this.

Thought 4: Alternatively, let me try to perform the division ... So there is a discrepancy in the coefficient of $p$.

Thought 5: Let me check the first approach again. ... Wait, but this seems conflicting with the previous result.

Thought 6: Wait, actually, no. The original polynomial is $x^9$ ... Let me check my calculations again.

Thought 7: First, let me verify the first approach where I substituted $\alpha$ into the polynomial. ... So the equations are correct.

Thought 8: But then why does the second approach give a different equation for the $x^2$ coefficient? Wait, maybe the second approach is not correct. ... This seems correct.

Thought 9: The second approach, however, might have an error in the expansion or reduction. Let me recheck ... there is a mistake in one of the approaches.

Thought 10: Wait, let me compute $\alpha^9 + p\alpha^6 + q\alpha^3 + r$ with $p = 6$, $q = 31$, $r = -1$ ... Hence, this approach might not be straightforward.

Thought 11: Alternatively, if I substitute y = x³, then P(x) = y³ + p y² + q y + r. But since x³ = 3x² -4x +1, then y = 3x² -4x +1. Therefore, P(x) = (3x² -4x +1)$^3$ + p(3x - " ...

AFTER ST (7413 OUTPUT TOKENS):
Thought 1: Okay, so I need to find the ordered triple $(p, q, r)$ such that the polynomial $x^3 - 3x^2 + 4x - 1$ is a factor of $x^9 + px^6 + qx^3 + r$. ... This seems complicated, but maybe there's a smarter way.

Thought 2: Alternatively, since the given polynomial is a factor, then all roots ... Let me check if indeed this is divisible by the cubic.

Thought 3: Alternatively, perhaps there is a better way. ... Let me try this approach.

Thought 4: Let me assume that $Q(x) = x^6 + ax^3 + b$. ... . **Final Answer** The ordered triple is $\boxed{(6, 31, -1)}$.

in the correct reasoning path, leading to inefficiency and potential error. This is the primary target for reduction by STPO.

The core objective of STPO is to mitigate **invalid switching**, thereby enhancing the persistence and efficiency of reasoning, without compromising the model's capacity for **valid switching**.

We use a simplified example to intuitively illustrate how ST works. Let $T$ denote a correct thought segment and $F$ denote an incorrect thought segment. As shown in Table 7, it compares a hypothetical reasoning chain for a correctly answered problem before and after applying Steady Thought training.

The example demonstrates that by penalizing the act of switching out of a correct thought ($T$), ST compels the model to **persist in the correct reasoning path** ($T_1 \rightarrow$ Correct), reducing 5 inefficient switches to just 1 efficient switch. Crucially, the model **retains the exploratory ability** to switch from an incorrect thought ($F_1$) to a correct thought ($T_1$). The overall reduction in effective switches

Table 7: Comparison of Reasoning Chains Before and After ST

| Stage | Response Chain | Total Switches | Invalid Switches | Valid Switches |
|-------|----------------|----------------|------------------|----------------|
| Before ST | $F_1 \rightarrow T_1 \rightarrow T_2 \rightarrow F_2 \rightarrow T_3 \rightarrow T_4 \rightarrow$ Correct | 5 | 3 | 2 |
| After ST | $F_1 \rightarrow T_1 \rightarrow$ Correct | 1 | 0 | 1 |

Table 8: Comparison of Number of Total Switches (NTS), Invalid Switches (NIS), and Valid Switches (NVS) generated by the model before and after applying ST.

| Method | MATH500 | | | AIME 2024 | | |
|--------|---------|-----|-----|-----------|-----|-----|
| | NTS | NIS | NVS | NTS | NIS | NVS |
| *DeepSeek-R1-Distill-Qwen-1.5B* | | | | | | |
| Vanilla | 3882 | 2130 | 1752 | 2849 | 412 | 2377 |
| Steady Thought | 2161(-44.3%) | 872(-59.1%) | 1289(-26.4%) | 4372(+53.5%) | 346(-16%) | 4026(+69.4%) |
| *Qwen3-8B* | | | | | | |
| Vanilla | 1928 | 1408 | 520 | 3934 | 1779 | 2155 |
| Steady Thought | 747(-61.3%) | 505(-64.1%) | 242(-53.5%) | 2189(-44.4%) | 854(-52.0%) | 1355(-37.1%) |

(from 2 to 1) is not due to impaired exploration, but because the model successfully persisted in the initial correct thought ($T_1$) and subsequently concluded the problem. This illustrates that STPO's benefit stems entirely from suppressing unwarranted abandonment, without compromising reasonable exploration.

The following experimental data, as shown in Table 8, quantitatively confirms the above mechanism. We compare the **Number of Total Switches (NTS)**, **Number of Invalid Switches (NIS, $T \rightarrow F$)**, and **Number of Valid Switches (NVS, $F \rightarrow T$)** for two different models before and after applying Steady Thought training.

For models across most datasets, the magnitude of the decrease in invalid switches is significantly larger than the decrease in valid switches. This directly proves that the resulting accuracy maintenance and token reduction primarily stem from the suppression of unwarranted abandonment ($T \rightarrow F$). When the weaker 1.5B model addresses the highly difficult AIME 2024 dataset, the number of valid switches increases by a remarkable **69.4%**. This crucial finding indicates that when the model encounters extremely challenging problems that necessitate extensive exploration, ST does not impede, but rather optimizes and enhances its reasonable exploration ability.

In summary, ST does not over-penalize exploration. Instead, it operates as an exploration-efficiency optimizer by rewarding persistence in the correct thought trajectory. This targeted optimization reduces the costly habit of unwarranted abandonment while preserving, and in challenging scenarios even improving, the model's ability to engage in necessary exploration.

## D FURTHER ANALYSIS ON ENTROPY THRESHOLDS

Due to the significant differences in scale and architecture among LLMs, it is reasonable to set an entropy threshold tailored to each model's characteristics. Therefore, we set different entropy thresholds for the 1.5B model and the 8B model, and this threshold proves to be robust across different task types for the same model. We provide specific experimental data to demonstrate this point in Table 9:

The experimental results indicate that when the threshold is reasonably set, high-quality data can be generated, enabling the model to effectively learn the thought-level optimization approach.

## E COMPUTATIONAL OVERHEAD OF DATA CONSTRUCTION

We provides a detailed discussion on the computational overhead of the Steady Thought pipeline, focusing on the cost incurred during the data construction stage (Thought Completion).

Table 9: Comparison of thought segmentation results under varying thresholds. NT: number of segmented thoughts.

| Base model | Threshold | NT | MATH500 | | AIME 2024 | | LiveCode | |
|---|---|---|---|---|---|---|---|---|
| | | | Acc(%)↑ | Tokens↓ | Acc(%)↑ | Tokens↓ | Acc(%)↑ | Tokens↓ |
| DeepSeek-R1-Distill-Qwen-1.5B | 2.8 | 20650 | 83.4 | 3854 | 29.2 | 9252 | **32.9** | 7427 |
| | **3.0** | 10444 | **84.4** | **2809** | **31.2** | **8606** | 32.4 | **7398** |
| | 3.2 | 4355 | 83.6 | 3657 | 28.3 | 10516 | 32.7 | 8913 |
| Qwen3-8B | 1.3 | 18792 | 93.8 | **2392** | 65.0 | 9048 | 76.0 | 6055 |
| | **1.5** | 11528 | **94.4** | 2869 | **65.8** | **8742** | **77.1** | **5759** |
| | 1.7 | 5706 | 94.2 | 3081 | 64.2 | 9331 | 74.3 | 5953 |

We calculated the total token consumption required for generating the preference data pairs during the Thought Completion stage for various model sizes as shown in Table 10.

It should be noted that for the 1.5B, 8B and 14B models, the Thought Completion stage utilized only **one A100 GPU** for inference acceleration. We consider this overhead to be acceptable and to yield a high return, particularly given its one-time nature and high efficiency.

Table 10: Token consumption overview for thought completion stage

| Model | Total Tokens Consumed | Average Tokens Consumed |
|---|---|---|
| DeepSeek-R1-Distill-Qwen-1.5B | 36,413,808 | 3,486 |
| Qwen3-8B | 82,056,359 | 7,058 |
| DeepSeek-R1-Distill-Qwen-14B | 59,722,645 | 3,448 |

# F QUANTITATIVE VALIDATION OF ENTROPY-BASED SEGMENTATION

We employed a Large Language Model (Gemini-2.5) with a specific Prompt to segment the reasoning Steps into Thoughts, treating the resulting segmentation as the ground truth. Comparing our entropy-based segmentation results against this standard, we achieved a precision of 85%. While entropy-based segmentation may introduce some noise, we still achieved comparatively good results, making it a highly cost-effective alternative to using LLMs for segmentation.

**Prompt for Thought Segmentation - Part1**

As a mathematical problem-solving analysis expert, please partition the following solution steps based on **Functional Shifts in Thinking**.

CORE PARTITIONING CRITERIA:

1. **Focus on Thought Function:** Each switch must signify a shift in the problem-solving goal or the core mathematical operation being performed. Even within the same major method, a pivot in key steps should be considered a switch.

2. **Operational Independence:** Logically connected steps that are pure, linear continuations should not be partitioned. However, functional transitions—such as moving from formula establishment to variable substitution, and then to result simplification—should be treated as switches.

3. **Insight and Bottleneck Response:** Any critical mathematical insight, non-trivial algebraic restructuring, or the establishment of a new analytical framework after complexity is encountered in the current method, shall be deemed a switch.

4. **Local Focus Principle:** Only consider whether the current step's strategy is the same as the strategy of the immediately preceding partition block.

5. **Depth of Analysis:** Maintain a deep analytical granularity.

   - Critical steps like formula establishment, model setting, or variable introduction should be partitioned independently.
   - Auxiliary steps such as verification, boundary condition checks, and result analysis must be partitioned separately.
   - In continuous derivations of the same type, a switch occurs when a new mathematical tool or theorem is introduced.

---

**Prompt for Thought Segmentation - Part2**

KEY IDENTIFICATION SIGNALS:

1. Methodological or Tool Transfer: For example, shifting from Geometric Relationship Analysis $\rightarrow$ Establishing Analytical Equations $\rightarrow$ Using Differential Calculus (Derivative Tools) for analysis.

2. Shift in Core Object of Study: For instance, moving from analyzing the properties of Point P $\rightarrow$ analyzing the extrema of function $\rightarrow$ analyzing the range of parameter $t$.

3. Appearance of Functional Transition Words: Focus on phrases like `"alternatively"` `"wait"` `"another approach"` `"let me check..."` which often signal a shift in thinking.

4. Critical Mathematical Restructuring: Such as performing non-obvious equivalent substitutions or changing the variable of integration.

5. Explicit Need for Check or Verification: A requirement to check a specific intermediate conclusion or boundary condition.

EXAMPLES:

- **Steps that can be considered a switch:**

    1. But let me think again if there's another way to approach this problem, maybe using algebraic manipulations instead of numerical examples.
    2. Another approach: Let me consider squaring the expression $z + 1/z$. Let me compute $(z + 1/z)^2 = z^2 + 2 + 1/z^2$. Therefore, $z^2 + 1/z^2 = (z + 1/z)^2 - 2$. Similarly, if I let $A = z + 1/z$, then $z^2 + 1/z^2 = A^2 - 2$. Therefore, the original expression $S = A + (A^2 - 2) = A^2 + A - 2$.
    3. So, maybe there is another rational root? Let me check $x =$ something else.

- **Steps that can't be considered a switch:**

    1. Therefore, if we want to write the same function with a different $c'$, then $c' = -b\phi' = -b[\phi + k(2\pi/b)] = -b\phi - 2\pi k$.
    2. Now, the next part is finding $\theta$. The formula for $\theta$ is the arctangent of $y$ over $x$, right? So $\theta = \arctan(y/x)$.

THE FOLLOWING ARE THE STEPS TO BE PARTITIONED:

OUTPUT FORMAT:

Please output the division results in JSON array format, containing no content other than JSON. The last `end_step` value must match the number of the final step:

`[[number of start_step, number of end_step],...]`

## G USE OF LARGE LANGUAGE MODELS IN MANUSCRIPT PREPARATION

This manuscript was prepared in compliance with ICLR 2026 policy, which includes disclosing the use of Large Language Models (LLMs) as an aid to improve the clarity and quality of the writing.

The LLM was employed for the following specific purposes:

- **Grammar and Spelling Correction:** To identify and rectify grammatical and typographical errors.
- **Clarity and Readability Enhancement:** To suggest alternative phrasings for improved sentence flow and readability.
- **Conciseness:** To assist in reducing verbosity and enhancing the directness of the writing.

It is important to note that the core intellectual content, including the research contributions, data analysis, and scientific claims, is solely attributable to the human authors. The use of LLMs was guided by rigorous oversight to ensure alignment with ethical academic practices.

