# OpenReview forum: "SteadyThought: Mitigating LLM Under-Thinking via Thought-Level Preference Optimization"
_ICLR.cc/2026/Conference — Submitted to ICLR 2026_

### Official Review · Reviewer_Knbc · 2025-10-21

**Soundness:** 2
**Presentation:** 2
**Contribution:** 2
**Rating:** 2
**Confidence:** 4

**Summary:**

This paper proposes Steady Thought (ST), a thought-level preference optimization framework designed to mitigate the under-thinking phenomenon in large reasoning models (LRMs).

**Strengths:**

Experimental results are good.

**Weaknesses:**

1. This work lacks significant novelty and does not offer compelling research insights. The three proposed components share similar ideas with prior studies, making the paper appear more like a compositional work built upon existing methods. Specifically, the designs in Sections 3.1 and 3.2 are rather straightforward, with many previous works employing analogous strategies; thus, the contribution in these parts can hardly be regarded as truly innovative and instead reflects a combination of existing tricks. Furthermore, the main idea of Section 3.3 is almost a direct extension of SimPO, with only minor adjustments in the level of application granularity. It lacks substantial algorithmic innovation and can be viewed as a mild variant rather than a new optimization framework. In summary, this paper represents an incremental improvement at the technical level—although the experimental results are satisfactory, the work as a whole resembles a technical report rather than a research study with strong originality;

2. While threshold tuning is discussed, the paper lacks qualitative or visual evidence (e.g., example trajectories) showing clearer reasoning stabilization;

3. Although the paper reports reduced token counts and small accuracy gains, these improvements could be attributed to shorter decoding or regularization effects rather than the claimed “thought-level optimization.” There is no qualitative or mechanistic evidence showing that the model truly learns a new form of reasoning persistence or gains interpretable control over its thought process.

**Questions:**

No more.

---

> ### Author Response · Authors · 2025-11-25
> **Response to Reviewer Knbc---Part 1/4**
>
> Dear Reviewer Knbc:
>
> Thank you for your valuable review, and for your high recognition of the experimental results presented in this paper.
>
> Your positive feedback is deeply encouraging and means a great deal to us. Below, we provide detailed responses to your insightful and constructive comments. We have incorporated the corresponding discussions into the appendix and added the necessary references in the main text. All changes are highlighted in blue.
>
> Throughout the discussion phase, we genuinely look forward to your feedback and are fully committed to addressing any remaining concerns. If our responses have properly resolved your questions, we would be deeply grateful if you would consider raising your score. If anything remains unclear or unconvincing, we would truly appreciate your further guidance, and we stand ready to engage promptly and constructively.
>
> >W1:This work lacks significant novelty and does not offer compelling research insights. The three proposed components share similar ideas with prior studies, making the paper appear more like a compositional work built upon existing methods. Specifically, the designs in Sections 3.1 and 3.2 are rather straightforward, with many previous works employing analogous strategies; thus, the contribution in these parts can hardly be regarded as truly innovative and instead reflects a combination of existing tricks. Furthermore, the main idea of Section 3.3 is almost a direct extension of SimPO, with only minor adjustments in the level of application granularity. It lacks substantial algorithmic innovation and can be viewed as a mild variant rather than a new optimization framework. In summary, this paper represents an incremental improvement at the technical level—although the experimental results are satisfactory, the work as a whole resembles a technical report rather than a research study with strong originality;
>
> Thank you for your valuable comments.
>
> **TL;DR:**
>
> We innovatively formalizes "Under-Thinking" as a preference optimization problem, introducing a novel framework that combines structured segmentation—integrating entropy with semantic pre-processing to ensure logical integrity—with logits-intervention for synthesizing ideal reasoning data. By elevating optimization to the Thought-Level, the method achieves a fundamental leap over previous coarse-grained approaches, enabling the model to selectively suppress unnecessary switching while preserving valuable exploration in long-chain reasoning.
>
> **Detailed Response:**
>
> We believe the core contribution of this study lies in the innovative formalization of the "Under-Thinking" phenomenon, and the systematic technical innovations across the three proposed stages (Segmentation, Completion, Optimization), which collectively form a novel framework with sufficient technical depth.
> First, the "Under-Thinking" phenomenon is frequent but has not received sufficient systematic attention. We innovatively formalize it as a preference optimization problem arising from the distinction in reasoning thought patterns. This fundamentally differs from prior heuristic methods focused on directly suppressing switching words, and we also innovatively propose a preference data synthesis method tailored for this problem.
>
> Second, the core innovation of the first stage, Thought Segmentation, lies in how we integrate entropy with structured pre-processing to overcome the critical limitations of solely relying on entropy in practical application, thereby ensuring the effectiveness and semantic integrity of the segments. We observed that purely relying on high-entropy points for segmentation often leads to semantically incomplete or logically broken units. This is particularly detrimental in reasoning tasks, and complete logical units are crucial for subsequent STPO learning. Our method is a two-step structured innovation, not mere entropy detection. First, we use a semi-structured delimiter to divide the response into Candidate Steps. This pre-processing step effectively prevents unreasonable breaks within formulas or code blocks caused by internal newlines, ensuring each Candidate Step is a relatively complete semantic unit. Subsequently, the application of the entropy threshold is not for direct segmentation but serves as a decision metric to judge whether this structured pre-processed Candidate Step should be considered the starting point of a new Thought. This combination ensures that the segmentation both captures the model's decision changes and maintains the semantic and logical completeness of each Thought unit.
>
> (to be continued...)

---

> ### Author Response · Authors · 2025-11-25
> **Response to Reviewer Knbc---Part 2/4**
>
> The second stage, Thought Completion, innovatively combines with the first stage and is designed to generate single-thought trajectories that lead directly to the answer (i.e., constructing the chosen samples). We use direct Logits intervention to force the model to persist in the current promising thought, synthesizing "ideal behavior" samples. These two stages effectively solve the core challenge of constructing high-quality, targeted learning signals, providing a unique and indispensable data foundation for the STPO framework.
>
> Existing methods aimed at suppressing excessive switching often optimize over an entire trajectory, failing to account for the necessary exploration contained within that trajectory. In the third stage, STPO, we innovatively elevate the optimization granularity to the Thought-Level, encouraging the model to persist in promising thoughts while selectively suppressing only unnecessary switching. This finer granularity of optimization enables the model to learn from "good persistence" and "bad switching," representing a fundamental leap over previous coarse-grained or purely suppressive methods.
>
> In conclusion, we believe the technical depth of our method is reflected in the combination of its multi-stage, high-quality data construction, and the elevation of the preference optimization granularity to the thought level, which together resolve the critical bottleneck of "Under-Thinking" in long-chain reasoning by current LLMs.

---

> ### Author Response · Authors · 2025-11-25
> **Response to Reviewer Knbc---Part 3/4**
>
> >W2:While threshold tuning is discussed, the paper lacks qualitative or visual evidence (e.g., example trajectories) showing clearer reasoning stabilization;
>
> Thank you for raising this important point.
>
> **TL;DR:**
>
> The qualitative trajectory comparison visually confirms thought stabilization. Before ST, the model exhibited "under-thinking" with 11 thoughts and 14,338 tokens, resulting in failure. After ST, the model achieved the correct answer with only 4 thoughts and 7,413 tokens. This drastic reduction in thought switching frequency and total tokens, while achieving success, provides clear evidence that ST creates a more stable and focused reasoning chain. We have updated this content into Table 6 in Appendix B and referenced it in Section 4.3.
>
> **Detailed Response:**
>
> We have added a representative case to Table 6 of the paper, which qualitatively showcases the change in the reasoning trajectory of the Qwen3-8B model on the same complex problem before and after applying the Steady Thought (ST) method, thereby visually proving the effect of thought stabilization achieved by our method.
>
>
> ### Table 6: Comparison of Reasoning Chains Before and After ST
>
> **Problem:**
> The polynomial $x^3 - 3x^2 + 4x - 1$ is a factor of $x^9 + px^6 + qx^3 + r.$ Enter the ordered triple $(p,q,r).$
>
> **Reference Answer:**
> (6, 31, -1)
>
> ---
> ### **Before ST** (14338 output tokens):
>
> **Thought 1:** Okay, so I need to find the ordered triple $(p, q, r)$ such that the polynomial $x^3 - 3x^2 + 4x - 1$ is a factor of $x^9 + p x^6 + q x^3 + r.$ ... Let me check if this makes sense.
>
> **Thought 2:** Wait, let me verify this result by another method to be sure. Maybe by polynomial division or by considering that $x^9 \dots$ So perhaps my earlier approach is correct.
>
> **Thought 3:** But let me check with the values I found. If $p = 6, \dots$ Let me check if $x^3 - 3x^2 + 4x - 1$ divides this.
>
> **Thought 4:** Alternatively, let me try to perform the division $ \dots$ So there is a discrepancy in the coefficient of $p$.
>
> **Thought 5:** Let me check the first approach again.$ \dots$ Wait, but this seems conflicting with the previous result.
>
> **Thought 6:** Wait, actually, no. The original polynomial is $x^9 \dots$ Let me check my calculations again.
>
> **Thought 7:** First, let me verify the first approach where I substituted $\alpha$ into the polynomial. $\dots$ So the equations are correct.
>
> **Thought 8:** But then why does the second approach give a different equation for the $x^2$ coefficient? Wait, maybe the second approach is not correct.$ \dots$ This seems correct.
>
> **Thought 9:** The second approach, however, might have an error in the expansion or reduction. Let me recheck $\dots$ there is a mistake in one of the approaches.
>
> **Thought 10:** Wait, let me compute $\alpha^9 + p \alpha^6 + q \alpha^3 + r$ with $p=6$, $q=31$, $r=-1$ $\dots$ Hence, this approach might not be straightforward.
>
> **Thought 11:** Alternatively, if I substitute y = x³, then P(x) = y³ + p y² + q y + r. But since x³ = 3x² -4x +1, then y = 3x² -4x +1. Therefore, P(x) = (3x² -4x +1)^3 + p(3x² -"$\dots$
>
> ---
> ### **After ST** (7413 output tokens):
>
> **Thought 1:** Okay, so I need to find the ordered triple $(p, q, r)$ such that the polynomial $x^3 - 3x^2 + 4x - 1$ is a factor of $x^9 + p x^6 + q x^3 + r.$ $\dots$ This seems complicated, but maybe there's a smarter way.
>
> **Thought 2:** Alternatively, since the given polynomial is a factor, then all roots $\dots$ Let me check if indeed this is divisible by the cubic.
>
> **Thought 3:** Alternatively, perhaps there is a better way. $\dots$ Let me try this approach.
>
> **Thought 4:** Let me assume that $Q(x) = x^6 + a x^3 + b. \dots$
> **Final Answer**
> The ordered triple is $\boxed{(6, 31, -1)}$.
>
> ---
>
> This case directly illustrates that, prior to applying ST, the model exhibited clear "under-thinking," characterized by frequently and unnecessarily switching thought (11 thoughts), leading to massive token waste and ultimately failing to provide a valid answer before exhausting resources.
>
> After applying ST, the model learned to persist in promising reasoning paths. The total thought count was sharply reduced to 4, total token consumption was nearly halved, and the model successfully and correctly solved the problem using a more stable and focused reasoning chain. This reduction in thought switching frequency and the successful delivery of the final answer provide clear, qualitative evidence that our method achieves reasoning stabilization.
>
> We have added a discussion regarding this part in Appendix B for the reviewer's reference.

---

> ### Author Response · Authors · 2025-11-25
> **Response to Reviewer Knbc---Part 4/4**
>
> >W3:Although the paper reports reduced token counts and small accuracy gains, these improvements could be attributed to shorter decoding or regularization effects rather than the claimed “thought-level optimization.” There is no qualitative or mechanistic evidence showing that the model truly learns a new form of reasoning persistence or gains interpretable control over its thought process.
>
> Thank you for raising this important point.
>
> **TL;DR:**
>
> Steady Thought (ST) proves genuine "thought-level optimization" by teaching the model selective persistence, not just shorter decoding. Mechanistic data shows ST drastically reduces Invalid Switching by up to 64.1%, while preserving the necessary Valid Switching. This selective inhibition demonstrates the model learns interpretable control over its reasoning path. We have added a discussion regarding this part in Appendix C and reference it in lines 378-379 of Section 4.4.2.
>
> **Detailed Response:**
>
> We agree that it is essential to demonstrate that the improvements stem from genuine control over the reasoning process rather than simple shorter decoding or regularization effects. Our arguments focus on the model's Thought Switching Control Ability and Reasoning Persistence, providing the following mechanistic evidence:
>
> Our proposed Steady Thought (ST) framework is not designed to simply "suppress all switching" (i.e., shorten decoding) but to train the model to perform rational and selective thought switching.
>
> First, we would like to clearly define the two types of switching behaviors we focus on:
>
> * **Valid Switching (Reasonable Exploration):** Switching from an incorrect thought ($F$) to the next thought, which represents the model undertaking necessary exploration.
> * **Invalid Switching (Unwarranted Abandonment):** Switching from a correct thought ($T$) to the next thought, which represents the model failing to identify and stick to the correct reasoning path.
>
> The core objective of STPO is to reduce the second type of "invalid switching," thereby improving the persistence and efficiency of reasoning, while retaining and even optimizing the ability for the first type of "valid switching."
>
> ---
>
> We use a simplified example to intuitively demonstrate the effect of STPO. Let $T$ be a correct thought and $F$ be an incorrect thought. The model's response to a correctly answered problem, before and after ST, is shown below.
>
> ### Table: Comparison of Reasoning Chains Before and After ST
> | Stage | Response Chain | Total Switches | Ineff. Switches | Eff. Switches |
> | :--- | :--- | :---: | :---: | :---: |
> | **Before ST** | $F_1 \to T_1 \to T_2 \to F_2 \to T_3 \to T_4 \to \text{Correct}$ | 5 | 3 | 2 |
> | **After ST** | $F_1 \to T_1 \to \text{Correct}$ | 1 | 0 | 1 |
>
> By penalizing the act of switching out of a correct thought, STPO makes the model learn to stick to the correct reasoning path, thereby reducing the inefficient 5 switches to an efficient 1 switch. Crucially, it still retains the model's exploratory ability to switch from an incorrect thought ($F$) to a correct thought ($T$). The reduction in the number of valid switches is not due to impaired exploration ability, but because the model has already persisted in the initial correct thought and outputted the answer. This demonstrates that the benefit of STPO stems entirely from reducing ineffective switching, without compromising reasonable exploration ability.
>
> ---
>
> Next, we provide specific experimental data to prove this point:
>
> Comparison of Number of Total Switches (NTS), Invalid Switches (NIS), and Valid Switches (NVS) generated by the model before and after applying ST.
>
> | Model | Status | NTS (MATH500) | NIS (MATH500) | NVS (MATH500) | NTS (AIME 2024) | NIS (AIME 2024) | NVS (AIME 2024) |
> | :--- | :--- | :---: | :---: | :---: | :---: | :---: | :---: |
> | **DeepSeek-R1-Distill-Qwen-1.5B** | Vanilla | 3882 | 2130 | 1752 | 2849 | 412 | 2377 |
> | | Steady Thought | 2161&nbsp;(-44.3%) | 872&nbsp;(-59.1%) | 1289&nbsp;(-26.4%) | 4372&nbsp;(+53.5%) | 346&nbsp;(-16.0%) | 4026&nbsp;(+69.4%) |
> | **Qwen3-8B** | Vanilla | 1928 | 1408 | 520 | 3934 | 1779 | 2155 |
> | | Steady Thought | 747&nbsp;(-61.3%) | 505&nbsp;(-64.1%) | 242&nbsp;(-53.5%) | 2189&nbsp;(-44.4%) | 854&nbsp;(-52.0%) | 1355&nbsp;(-37.1%) |
>
> Across all experiments, the magnitude of the decrease in invalid switches is **significantly larger** than the decrease in valid switches. This proves that the maintenance of accuracy and the reduction in token count primarily result from the suppression of unwarranted abandonment.
>
> This mechanism of selective inhibition demonstrates that the model has learned to evaluate the value of its current thought. It adheres to a promising reasoning path and only engages in exploratory switching when the current path's value is low. This represents a clear instance of thought-level control learning, not merely a generalized regularization or overall reduction in decoding length.

---

### Official Review · Reviewer_QcuC · 2025-10-31

**Soundness:** 2
**Presentation:** 2
**Contribution:** 2
**Rating:** 4
**Confidence:** 3

**Summary:**

The authors observe that models often discover a correct reasoning path early in inference but then perform numerous unnecessary thought switches, undermining reasoning depth and coherence. To address this "under-thinking" problem, they propose Steady Thought (ST), a framework that: (1) segments thoughts using entropy-based detection, (2) completes each thought without further switching, and (3) constructs thought-level preference pairs based on final correctness. Experimental results show that ST successfully mitigates under-thinking by reducing unnecessary switches, leading to more focused reasoning while maintaining or even enhancing performance.

**Strengths:**

- Clearly identifies and formalizes "under-thinking" as a thought-level preference learning problem, more fine-grained than prior response-level approaches, preserving the model's flexibility to explore alternative reasoning paths when needed.
- The proposed ST framework combining entropy-based thought segmentation with a SimPO-inspired preference objective (STPO) that effectively mitigates length bias.
- Strong empirical results: consistent accuracy gains and token reductions across multiple models and datasets, including out-of-distribution generalization to code tasks despite training only on math data.

**Weaknesses:**

- The reliance on predefined switch tokens (e.g., "wait", "alternatively") limits generalization, especially for models or domains that switch thoughts implicitly without explicit lexical cues. In contrast, concurrent work like SwiReasoning (arXiv:2510.05069) effectively handles implicit thought switching in latent space.
- Thought segmentation hinges on a tunable entropy threshold; while ablations are provided, its robustness across diverse reasoning styles or model architectures remains unclear.
- The computational overhead of the ST pipeline, particularly completion per response during data construction, is not adequately discussed.
- The method assumes a correct answer can be derived by completing a single early thought, which may not hold for problems requiring genuine multi-stage exploration or backtracking.
- The experimental evaluation is limited in scope: only two models and three benchmarks are tested. The paper does not investigate whether the approach scales effectively to larger reasoning models.

**Questions:**

See the Weaknesses above.

---

> ### Author Response · Authors · 2025-11-24
> **Response to Reviewer QcuC---Part 1/5**
>
> Dear Reviewer QcuC:
>
> Thank you for your valuable review, and for your high recognition of the problem definition, methodology, and experimental results presented in this paper. Your positive feedback is deeply encouraging and means a great deal to us. Below, we provide detailed responses to your insightful and constructive comments. We have revised the corresponding content in the main text or added new discussions in the appendix. When modified in the main text, the changes are marked in blue; when an entire new section is added in the appendix, the title is changed to blue.
>
> Throughout the discussion phase, we genuinely look forward to your feedback and are fully committed to addressing any remaining concerns. If our responses have properly resolved your questions, we would be deeply grateful if you would consider raising your score. If anything remains unclear or unconvincing, we would truly appreciate your further guidance, and we stand ready to engage promptly and constructively.
>
> >W1:The reliance on predefined switch tokens (e.g., "wait", "alternatively") limits generalization, especially for models or domains that switch thoughts implicitly without explicit lexical cues. In contrast, concurrent work like SwiReasoning (arXiv:2510.05069) effectively handles implicit thought switching in latent space.
>
> Thank you for raising this important point.
>
> **TL;DR:**
>
> ST segments thoughts based on high latent uncertainty. Suppressing switching words is only a data construction method to generate high-quality, non-redundant training samples. STPO learns from these samples, enabling the model to internalize persistence on promising paths, ultimately reducing ineffective switching.
>
> **Detailed Response:**
>
> We fully agree that models may exhibit implicit thought switching across different domains and scales. However, our method is consistent and complementary to the core idea of SwiReasoning: we also recognize and leverage uncertainty in the latent space as a signal for thought switching.
> Our method does not set the learning objective as suppressing these explicit words, but rather uses them as a means of data construction.
>
> Our entire framework is aligned with SwiReasoning's view, recognizing and utilizing high uncertainty (high entropy) in the latent space as a signal indicating that the model is about to explore multiple lines of thought. In the first stage, we use a preset entropy threshold and entropy detection to segment the response into complete Thoughts, which is essentially partitioning at the uncertainty peaks in the latent space. This is consistent with methods for identifying switching in the latent space.
>
> The suppression of predefined switching words is not intended to teach the model to avoid these explicit terms. It is merely a way to construct high-quality training data. Through this heuristic intervention, we force the model to persist on a thought identified as promising, thereby generating a high-quality trajectory without explicit thought switching (which serves as the positive sample for STPO).
>
> By learning the preference difference between chosen samples and rejected samples, the model autonomously learns to stick to a promising path in the latent space. The result of this learning is reflected in the reduction of invalid switching counts in the explicit space.

---

> ### Author Response · Authors · 2025-11-24
> **Response to Reviewer QcuC---Part 2/5**
>
> >W2:Thought segmentation hinges on a tunable entropy threshold; while ablations are provided, its robustness across diverse reasoning styles or model architectures remains unclear.
>
> Thank you for raising this important point.
>
> **TL;DR:**
>
> Thresholds are Model-Specific but Task-Robust. Due to architectural differences, we use distinct optimal entropy thresholds for the 1.5B (3.0) and 8B (1.5) models. Experiments confirm that this model-specific optimal threshold is highly robust across diverse task types (Math and Code), consistently maximizing accuracy and minimizing token usage, which ensures high-quality thought segmentation data.We discuss threshold settings on more models in Appendix D and reference it in Section 4.4.3.
>
> **Detailed Response:**
>
> First, we want to clarify that due to the significant differences in scale and architecture among LLMs, it is reasonable to set an entropy threshold tailored to each model's characteristics. Therefore, we set different entropy thresholds for the 1.5B model and the 8B model, and this threshold proves to be robust across different task types for the same model.
> We now provide specific experimental data to demonstrate this point:
>
> ### Comparison of thought segmentation results under varying thresholds.
>  **NT**: number of segmented thoughts.
>
> | Base model | Threshold | NT | MATH500 (Acc(\%)$\uparrow$) | MATH500 (Tokens$\downarrow$) | AIME 2024 (Acc(\%)$\uparrow$) | AIME 2024 (Tokens$\downarrow$) | LiveCode (Acc(\%)$\uparrow$) | LiveCode (Tokens$\downarrow$) |
> | :--- | :---: | :---: | :---: | :---: | :---: | :---: | :---: | :---: |
> | DeepSeek-R1-Distill-Qwen-1.5B | 2.8 | 20650 | 83.4 | 3854 | 29.2 | 9252 | **32.9** | 7427 |
> | | **3.0** | 10444 | **84.4** | **2809** | **31.2** | **8606** | 32.4 | **7398** |
> | | 3.2 | 4355 | 83.6 | 3657 | 28.3 | 10516 | 32.7 | 8913 |
> | Qwen3-8B | 1.3 | 18792 | 93.8 | **2392** | 65.0 | 9048 | 76.0 | 6055 |
> | | **1.5** | 11528 | **94.4** | 2869 | **65.8** | **8742** | **77.1** | **5759** |
> | | 1.7 | 5706 | 94.2 | 3081 | 64.2 | 9331 | 74.3 | 5953 |
>
> The experimental results indicate that when the threshold is reasonably set, high-quality data can be generated, enabling the model to effectively learn the thought-level optimization approach.

---

> ### Author Response · Authors · 2025-11-24
> **Response to Reviewer QcuC---Part 3/5**
>
> >W3:The computational overhead of the ST pipeline, particularly completion per response during data construction, is not adequately discussed.
>
> Thank you for raising this important point.
>
> **TL;DR:**
>
> The computational overhead of the Thought Completion stage is acceptable and yields a high return. It is a one-time data synthesis step with a moderate total token consumption (e.g., 1.5B model uses ~36M tokens on one A100 GPU), which produces high-value preference data essential for STPO training. We discuss the consumption generated by thought completion in Appendix E and reference it in Section 3.2.
>
> **Detailed Response:**
>
> Thank you for your concern regarding the computational overhead of the Steady Thought Pipeline, especially the cost incurred during the data construction stage (Thought Completion). We consider this overhead to be **acceptable** and to yield a **high return**, particularly given its one-time nature and high efficiency.
>
> Below is a detailed discussion of the computational overhead for the Thought Completion stage:
>
> We have calculated the total token consumption for generating preference data pairs during the Thought Completion stage:
>
> | Model Size | Total tokens consumed | Average tokens consumed  |
> | :--- | :---: | :---: |
> | 1.5B | 36,413,808 | 3,486 |
> | 8B | 82,056,359 | 7,058 |
> | 14B | 59,722,645 | 3,448 |
>
> It should be noted that for the 1.5B and 8B models, the Thought Completion stage utilized only **one A100 GPU** for inference acceleration.
>
> We believe the overhead of the Thought Completion stage is reasonable because it is a **one-time, high-value data synthesis step**, whose return is the high-quality, hard-to-obtain preference data used for STPO training.

---

> ### Author Response · Authors · 2025-11-24
> **Response to Reviewer QcuC---Part 4/5**
>
> >W4:The method assumes a correct answer can be derived by completing a single early thought, which may not hold for problems requiring genuine multi-stage exploration or backtracking.
>
> Thank you for raising this important point.
>
> **TL;DR:**
>
> The "solvable by a single early thought" assumption in the Thought Completion stage is an engineering choice to provide ST with the strongest signal for persistence ("persistence leads to success"). This idealized data does not constrain the model's ability to handle complex problems. Empirical results (e.g., 69.4% increase in valid switches for the 1.5B model on AIME 24) prove that ST successfully generalizes this signal into a universal strategy, allowing the model to perform multi-stage exploration and necessary switching with greater discernment during inference.We further discuss the impact of ST on the model's exploration capability and provide specific experimental data in Appendix C, and reference it in Section 4.4.2.
>
> **Detailed Response:**
>
> We agree that the Thought Completion stage, by forcing the completion of an early thought to construct training data, does introduce the idealized assumption of "solvable by a single thought." However, this is an engineering choice designed to provide the STPO framework with the strongest possible learning signal for persisting in a promising thought.
>
> Our core view is that this idealized data construction does not impose a "single thought" constraint during the training phase; its learning objective is a general strategy, not a specific path. The central goal of ST is to teach the model to maintain belief in a high-potential path when facing uncertainty, suppressing unnecessary and wasteful switching, while preserving and optimizing its ability for necessary exploration and backtracking in complex problems.
>
> The "single thought successful path" data generated during the Thought Completion stage demonstrates the ideal behavioral pattern to the model: "If you persist, you can succeed." ST internalizes this pattern as a general strategy.
>
> Table: Comparison of Number of Total Switches (NTS), Invalid Switches (NIS), and Valid Switches (NVS) generated by the model before and after applying ST.
>
> | Model | Status | NTS (MATH500) | NIS (MATH500) | NVS (MATH500) | NTS (AIME 2024) | NIS (AIME 2024) | NVS (AIME 2024) |
> | :--- | :--- | :---: | :---: | :---: | :---: | :---: | :---: |
> | **DeepSeek-R1-Distill-Qwen-1.5B** | Vanilla | 3882 | 2130 | 1752 | 2849 | 412 | 2377 |
> | | Steady Thought | 2161&nbsp;(-44.3%) | 872&nbsp;(-59.1%) | 1289&nbsp;(-26.4%) | 4372&nbsp;(+53.5%) | 346&nbsp;(-16.0%) | 4026&nbsp;(+69.4%) |
> | **Qwen3-8B** | Vanilla | 1928 | 1408 | 520 | 3934 | 1779 | 2155 |
> | | Steady Thought | 747&nbsp;(-61.3%) | 505&nbsp;(-64.1%) | 242&nbsp;(-53.5%) | 2189&nbsp;(-44.4%) | 854&nbsp;(-52.0%) | 1355&nbsp;(-37.1%) |
>
> Empirical results confirm this generalization ability: for complex problems like AIME 24, which require genuine multi-stage exploration and backtracking, our model did not reduce its effective exploration count; in fact, the number of effective switches even increased for the 1.5B model, and the final accuracy was actually improved. This indicates that the ST framework successfully generalized the ideal pattern from the training data into a universal strategy. During actual inference, the model is still capable of multi-stage exploration and strategy switching when necessary, but it operates with greater discernment, avoiding the wasted exploration caused by "under-thinking."
>
> Therefore, the idealized assumption of the Thought Completion stage is a tool for efficiently constructing the training signal, but it does not impair the model's ability to handle complex multi-stage problems.

---

> ### Author Response · Authors · 2025-11-24
> **Response to Reviewer QcuC---Part 5/5**
>
> >W5:The experimental evaluation is limited in scope: only two models and three benchmarks are tested. The paper does not investigate whether the approach scales effectively to larger reasoning models.
>
> Thank you for raising this important point.
>
> **TL;DR:**
>
> We have added a 14B-sized model and included the performance of all models on the GSM8K dataset. We have updated Table 1 in the paper and revised the description of Table 1 in lines 285-298, as well as added a new description of the GSM8K dataset in lines 257-261.
>
>
> **Detailed Response:**
>
> We have added the GSM8K dataset and supplemented the results of the 14B model on various datasets. We present these alongside the existing data in the main table. It can be seen that the new 14B model shows good optimization across all benchmarks after applying ST (Steady Thought).
>
> ### Experimental results on three large reasoning models
>
> #### DeepSeek-R1-Distill-Qwen-1.5B
> | Method | MATH-500 Acc(%) | MATH-500 Tokens | AIME 2024 Acc(%) | AIME 2024 Tokens | GSM8K Acc(%) | GSM8K Tokens | LiveCode Acc(%) | LiveCode Tokens | Overall Acc(%) | Overall Tokens |
> | :--- | :---: | :---: | :---: | :---: | :---: | :---: | :---: | :---: | :---: | :---: |
> | **Vanilla** | 82.0 | 4385 | 27.5 | 11273 | 81.9 | 1448 | 30.3 | 9623 | 55.43 | 6682 |
> | **NoThink** | 65.8 | 749 | 8.7 | 3185 | 53.6 | 263 | 20.7 | 813 | 37.20 (-18.23) | 1252 (-81.3%) |
> | **NOWAIT** | 80.6 | 2433 | 20.8 | 7000 | 66.1 | 2078 | 28.3 | 4927 | 48.95 (-6.48) | 4109 (-38.5%) |
> | **SEAL** | 82.6 | 3252 | 25.4 | 9120 | 79.7 | 860 | 29.5 | 7948 | 54.30 (-1.13) | 5295 (-20.8%) |
> | **Steady Thought** | 84.4 | 2809 | 31.2 | 8606 | 81.3 | 1254 | 32.4 | 7398 | 57.33 (+1.9) | 5016 (-24.9%) |
>
> #### Qwen3-8B
> | Method | MATH-500 Acc(%) | MATH-500 Tokens | AIME 2024 Acc(%) | AIME 2024 Tokens | GSM8K Acc(%) | GSM8K Tokens | LiveCode Acc(%) | LiveCode Tokens | Overall Acc(%) | Overall Tokens |
> | :--- | :---: | :---: | :---: | :---: | :---: | :---: | :---: | :---: | :---: | :---: |
> | **Vanilla** | 91.4 | 4724 | 62.1 | 10895 | 95.6 | 1759 | 71.8 | 7112 | 80.23 | 6122 |
> | **NoThink** | 85.2 | 933 | 25.8 | 3504 | 93.6 | 289 | 45.6 | 584 | 62.55 (-17.68) | 1327 (-78.3%) |
> | **NOWAIT** | 61.0 | 13274 | 26.3 | 14333 | 73.3 | 12369 | 75.5 | 5226 | 59.03 (-21.20) | 11300 (+84.6%) |
> | **SEAL** | 92.2 | 4034 | 58.8 | 10372 | 95.9 | 1421 | 83.4 | 6414 | 82.58 (+2.35) | 6940 (-8.4%) |
> | **Steady Thought** | 94.4 | 2869 | 65.8 | 8742 | 96.1 | 862 | 77.1 | 5759 | 83.35 (+3.12) | 4558 (-25.5%) |
>
> #### DeepSeek-R1-Distill-Qwen-14B
> | Method | MATH-500 Acc(%) | MATH-500 Tokens | AIME 2024 Acc(%) | AIME 2024 Tokens | GSM8K Acc(%) | GSM8K Tokens | LiveCode Acc(%) | LiveCode Tokens | Overall Acc(%) | Overall Tokens |
> | :--- | :---: | :---: | :---: | :---: | :---: | :---: | :---: | :---: | :---: | :---: |
> | **Vanilla** | 93.6 | 3349 | 60.4 | 8974 | 94.8 | 894 | 70.1 | 6789 | 79.73 | 5001 |
> | **NoThink** | 41.7 | 824 | 27.1 | 3279 | 90.1 | 256 | 44.0 | 708 | 50.73 (-29.00) | 1266 (-74.7%) |
> | **NOWAIT** | 75.6 | 3314 | 33.8 | 9431 | 86.3 | 936 | 64.3 | 5099 | 65.00 (-14.73) | 4695 (-6.1%) |
> | **SEAL** | 92.6 | 3253 | 60.8 | 8831 | 94.7 | 880 | 75.1 | 6706 | 80.80 (+1.07) | 4917 (-1.7%) |
> | **Steady Thought** | 94.2 | 2455 | 65.4 | 7554 | 95.1 | 715 | 74.3 | 5825 | 82.25 (+2.52) | 4137 (-17.3%) |

---

### Official Review · Reviewer_nhq2 · 2025-11-01

**Soundness:** 2
**Presentation:** 3
**Contribution:** 2
**Rating:** 4
**Confidence:** 3

**Summary:**

This paper, “SteadyThought: Mitigating LLM Under-Thinking via Thought-Level Preference Optimization,” addresses the phenomenon of under-thinking in large reasoning models (LRMs)—a tendency to switch reasoning trajectories excessively, abandoning promising thoughts prematurely. To solve this, they propose Steady Thought (ST), which consists of thought segmentation, thought completion, and fine-grained preference optimization. Experiments on multiple reasoning models across datasets demonstrate the effectiveness of the proposed approach.

**Strengths:**

(1) This paper addresses the frequent thought switching problem by Steady Thought, a thought-level preference optimization framework.

(2) Thought segmentation and thought completion are used to construct preference pairs to optimize LLMs.

(3) Experiments are tested on two large reasoning models across three datasets.

**Weaknesses:**

(1) The reasonability of using entropy to segment thoughts is not well justified.

(2) The technical depth and novelty of the proposed method is somewhat limited.

(3) The results in Table 2 are questionable. The percentage of correct thoughts is reduced when using steady thought. To my understanding, steady thought should reduce the number of thoughts but increase the percentage of correct thoughts.

**Questions:**

Is there any quantitative metric to show the effectiveness of using entropy to segment thoughts?

---

> ### Author Response · Authors · 2025-11-25
> **Response to Reviewer nhq2---Part 1/4**
>
> Dear Reviewer nhq2:
>
> Thank you for your valuable review, and for your high recognition of the scientific problem, data construction method, and experimental results presented in this paper. Your positive feedback is deeply encouraging and means a great deal to us. Below, we provide detailed responses to your insightful and constructive comments. We have incorporated the corresponding discussions into the appendix and added the necessary references in the main text. All changes are highlighted in blue.
>
> Throughout the discussion phase, we genuinely look forward to your feedback and are fully committed to addressing any remaining concerns. If our responses have properly resolved your questions, we would be deeply grateful if you would consider raising your score. If anything remains unclear or unconvincing, we would truly appreciate your further guidance, and we stand ready to engage promptly and constructively.
>
> >W1:The reasonability of using entropy to segment thoughts is not well justified.
>
> Thank you for your valuable comments.
>
> **TL;DR:**
>
> We mitigating "under-thinking" by framing it as a thought-level preference optimization problem. It uses token-level entropy peaks to segment an LLM's reasoning process. These high-uncertainty points signify critical shifts in the model's strategy (e.g., switching solution thoughts). By detecting these peaks, the method identifies the natural boundaries between distinct "thoughts," which are then used to construct preference data pairs for optimization.
>
> **Detailed Response:**
>
> We formalize the problem of "under-thinking" as a thought-level preference optimization problem, thus requiring the construction of preference data pairs distinguishable at the thought level. In the auto-regressive generation process, the token-level entropy $$H(p_t) = -\sum_i p_i \log p_i$$ quantifies the model's uncertainty in selecting the next token. Many existing works suggest that a high entropy value indicates that the model is facing a critical decision point of beginning to explore different lines of thought. This is because when the model faces a significant change in strategy or path (e.g., switching from a failed attempt to a new mathematical formula or programming method), it must weigh multiple seemingly plausible subsequent steps. At this point, the model's output vocabulary probability distribution tends to become uniform, leading the token-level entropy to reach a local peak. Therefore, using entropy as the basis for thought segmentation is grounded in an understanding of the LLM's internal reasoning mechanism: the internal thought switching within the model (i.e., the re-selection of strategy) manifests as a state of high uncertainty during the generation process. We capture these high-uncertainty peaks using an entropy threshold, thereby delineating the natural boundaries of thoughts.
>
> >Q1:Is there any quantitative metric to show the effectiveness of using entropy to segment thoughts?
>
> **Detailed Response:**
>
> We employed a Large Language Model (Gemini-2.5) with a specific Prompt to segment the reasoning Steps into Thoughts, treating the resulting segmentation as the ground truth. Comparing our entropy-based segmentation results against this standard, we achieved a precision of **85%**. While entropy-based segmentation may introduce some noise, we still achieved comparatively good results, making it a highly cost-effective alternative to using LLMs for segmentation. We have added the relevant results and the prompt for LLM to Appendix F.

---

> ### Author Response · Authors · 2025-11-25
> **Response to Reviewer nhq2---Part 2/4**
>
> >W2:The technical depth and novelty of the proposed method is somewhat limited.
>
> Thank you for raising this important point.
>
> **TL;DR:**
>
> The core technical depth lies in the three-stage innovative framework solving the LLM “under-thinking” problem:
>
> Novel Formalization: Formalizing "under-thinking" as a thought-level preference optimization problem.
>
> Data Construction (Segmentation & Completion): Using structured segmentation for semantic integrity; employing Logits intervention to efficiently synthesize high-quality "persistence" training signals (Chosen samples).
>
> Optimization Upgrade (STPO): Elevating optimization granularity to the Thought-Level, enabling the model to internalize persistence on promising paths and effectively reduce ineffective switching.
>
> **Detailed Response:**
>
> We believe the core contribution of this study lies in the innovative formalization of the "Under-Thinking" phenomenon, and the systematic technical innovations across the three proposed stages (Segmentation, Completion, Optimization), which collectively form a novel framework with sufficient technical depth.
>
> First, the "Under-Thinking" phenomenon is frequent but has not received sufficient systematic attention. We innovatively formalize it as a preference optimization problem arising from the distinction in reasoning thought patterns. This fundamentally differs from prior heuristic methods focused on directly suppressing switching words, and we also innovatively propose a preference data synthesis method tailored for this problem.
>
> Second, the core innovation of the first stage, Thought Segmentation, lies in how we integrate entropy with structured pre-processing to overcome the critical limitations of solely relying on entropy in practical application, thereby ensuring the effectiveness and semantic integrity of the segments. We observed that purely relying on high-entropy points for segmentation often leads to semantically incomplete or logically broken units. This is particularly detrimental in reasoning tasks, and complete logical units are crucial for subsequent STPO learning. Our method is a two-step structured innovation, not mere entropy detection. First, we use a semi-structured delimiter (.\n\n) to divide the response into Candidate Steps. This pre-processing step effectively prevents unreasonable breaks within formulas or code blocks caused by internal newlines, ensuring each Candidate Step is a relatively complete semantic unit. Subsequently, the application of the entropy threshold is not for direct segmentation but serves as a decision metric to judge whether this structured pre-processed Candidate Step should be considered the starting point of a new Thought. This combination ensures that the segmentation both captures the model's decision changes and maintains the semantic and logical completeness of each Thought unit.
>
> The second stage, Thought Completion, innovatively combines with the first stage and is designed to generate single-thought trajectories that lead directly to the answer (i.e., constructing the chosen samples). We use direct Logits intervention to force the model to persist in the current promising thought, synthesizing "ideal behavior" samples. These two stages effectively solve the core challenge of constructing high-quality learning signals that are discriminative at the thought level, thereby providing a unique and indispensable data foundation for the STPO.
>
> Existing methods aimed at suppressing excessive switching often optimize over an entire trajectory, failing to account for the necessary exploration contained within that trajectory. In the third stage, STPO, we innovatively elevate the optimization granularity to the Thought-Level. This encourages the model to persist in promising thoughts while selectively suppressing only unnecessary switching. This finer granularity of optimization enables the model to learn from  "good persistence" and "bad switching," representing a fundamental leap over previous coarse-grained or purely suppressive methods.
>
> In conclusion, we believe the technical depth of our method is reflected in the combination of its multi-stage, high-quality data construction, and the elevation of the preference optimization granularity to the thought level, which together resolve the critical bottleneck of "Under-Thinking" in long-chain reasoning by current LLMs.

---

> ### Author Response · Authors · 2025-11-25
> **Response to Reviewer nhq2---Part 3/4**
>
> >W3:The results in Table 2 are questionable. The percentage of correct thoughts is reduced when using steady thought. To my understanding, steady thought should reduce the number of thoughts but increase the percentage of correct thoughts.
>
> Thank you for your valuable question.
>
> **TL;DR:**
>
> The reviewer's concern about the "decrease in the percentage of correct thoughts" validates STPO's success: the goal is to significantly reduce Invalid Switching (abandoning correct thought). Since the count of correct thoughts equals Invalid Switches, the percentage drop means STPO successfully suppressed unwarranted abandonment, leading to improved reasoning efficiency by persisting in early correct thoughts, while maintaining accuracy.We have added further explanations regarding the implications of Table 2 in lines 368-373 of Section 4.4.2 in the paper. We further discuss the impact of ST on the model's output and provide specific experimental data in Appendix C, and reference it in lines 378-379 of Section 4.4.2.
>
> **Detailed Response:**
>
> We apologize for the initial lack of clarity in our description, which led to this misunderstanding. We will clarify the rationale behind Table 2 by redefining our optimization objective and providing a more detailed analysis. The core goal of ST is not to indiscriminately suppress all thought switches, but to mitigate the "Under-Thinking" problem. To this end, we clearly define two types of switching behaviors:
>
> Valid Switching (Reasonable Exploration): Switching from an incorrect thought (F) to the next thought (X), which represents the model undertaking necessary exploration.
>
> Invalid Switching (Unwarranted Abandonment): Switching from a correct thought (T) to the next thought (X), which represents the model failing to identify and stick to the correct reasoning path.
> The primary objective of ST is to significantly reduce the second type, "Invalid Switching," thereby enhancing the model's persistence in the correct path and its reasoning efficiency.
>
> Your observation that "the total number of thoughts decreased, but the percentage of correct thoughts decreased" is precisely a manifestation of ST successfully achieving its goal:
>
> Reduction in Total Thoughts: ST penalized unnecessary switching, causing the model to persist longer on the correct thought until the task is complete, leading to a significant drop in the total number of thoughts (i.e., improved efficiency).
>
> Reduction in Percentage of Correct Thoughts: The count of "correct thoughts" in Table 2 is equal to the count of Invalid Switches. The reduction in the percentage of correct thoughts indicates that the decrease in Invalid Switches is substantially greater than the decrease in Valid Switches. Since the model's overall accuracy is maintained, the reduction in the number of Valid Switches is not due to impaired exploration ability, but because the model persisted on an earlier correct thought and outputted the answer.
>
> We use a simplified example to intuitively demonstrate the effect of ST:
>
> Let T be a correct thought and F be an incorrect thought. The model's response to a correctly answered problem, before and after ST , is as follows:
>
> ### Table: Comparison of Reasoning Chains Before and After ST
> | Stage | Response Chain | Total Switches | Ineff. Switches | Eff. Switches |
> | :--- | :--- | :---: | :---: | :---: |
> | **Before ST** | $F_1 \to T_1 \to T_2 \to F_2 \to T_3 \to T_4 \to \text{Correct}$ | 5 | 3 | 2 |
> | **After ST** | $F_1 \to T_1 \to \text{Correct}$ | 1 | 0 | 1 |
>
> By penalizing the T $\to$ X behavior, ST reduces the invalid 5 switches (3 of which were unwarranted abandonments) to an valid 1 switch. This drastically reduces the count of "Correct Thoughts (Invalid Switches)," which consequently leads to the reduction in the "percentage of correct thoughts" in Table 2.
>
> （to be continued...）

---

> ### Author Response · Authors · 2025-11-25
> **Response to Reviewer nhq2---Part 4/4**
>
> ### Table: Comparison of Number of Total Switches (NTS), Invalid Switches (NIS), and Valid Switches (NVS) generated by the model before and after applying ST.
>
> | Model | Status | NTS (MATH500) | NIS (MATH500) | NVS (MATH500) | NTS (AIME 2024) | NIS (AIME 2024) | NVS (AIME 2024) |
> | :--- | :--- | :---: | :---: | :---: | :---: | :---: | :---: |
> | **DeepSeek-R1-Distill-Qwen-1.5B** | Vanilla | 3882 | 2130 | 1752 | 2849 | 412 | 2377 |
> | | Steady Thought | 2161&nbsp;(-44.3%) | 872&nbsp;(-59.1%) | 1289&nbsp;(-26.4%) | 4372&nbsp;(+53.5%) | 346&nbsp;(-16.0%) | 4026&nbsp;(+69.4%) |
> | **Qwen3-8B** | Vanilla | 1928 | 1408 | 520 | 3934 | 1779 | 2155 |
> | | Steady Thought | 747&nbsp;(-61.3%) | 505&nbsp;(-64.1%) | 242&nbsp;(-53.5%) | 2189&nbsp;(-44.4%) | 854&nbsp;(-52.0%) | 1355&nbsp;(-37.1%) |
>
> To more clearly prove that we successfully reduced invalid switching and improved reasoning efficiency, our detailed analysis of switching counts across different models and tasks shows that in all cases, the number of Invalid Switches achieved a significant reduction (up to 64.1% decrease). This directly proves that STPO successfully inhibited the model's tendency to prematurely abandon the correct reasoning path.
>
> In summary, the results in Table 2, along with our decomposition of the switching counts, clearly support our core objective: to enhance the model's reasoning persistence and efficiency by substantially reducing unnecessary ineffective switching (which consequently lowers the percentage of correct thoughts), while maintaining or even optimizing its exploratory capabilities.

---

### Official Review · Reviewer_YpkJ · 2025-11-02

**Soundness:** 3
**Presentation:** 3
**Contribution:** 2
**Rating:** 6
**Confidence:** 4

**Summary:**

The paper proposes a novel framework called Steady Thought, which aims to mitigate the pervasive phenomenon of "under-thinking" in Large Reasoning Models during complex reasoning tasks. This phenomenon is characterized by the model's failure to persevere and fully explore a promising reasoning path, instead switching excessively and inefficiently between thought trajectories.

**Strengths:**

1.The writing is clear, and the motivation is well articulated.

2.Recognizing the sensitivity of DPO to length bias, the authors introduce a length-normalized STPO objective based on SimPO, which is crucial for their method since the rejected switching trajectories are typically much longer than the selected completions.

3.Both accuracy and efficiency are improved.

**Weaknesses:**

1.The paper states that STPO reduces the number of tokens. For very challenging problems such as AIME 2024, the model may need multiple switches to find the correct reasoning path, indicating that exploration is valuable. Does ST risk over-penalizing reasonable exploration and switching, and to what extent might this affect the model’s ability to initially explore diverse reasoning strategies?

2.The core preprocessing step, thought segmentation, relies on entropy-based detection and predefined thresholds. Although the authors mention hyperparameter tuning, there is a lack of analysis on the stability and robustness of these thresholds across different model scales (1.5B vs. 8B) and task types (mathematics vs. programming).

3.In the thought completion stage, the model prevents switching by directly lowering the logits of trigger words such as “wait” and “alternatively.” This heuristic intervention might contradict the goal of STPO, which aims to implicitly suppress such words through learning.

**Questions:**

see weaknesses.

---

> ### Author Response · Authors · 2025-11-24
> **Response to Reviewer YpkJ---Part 1/4**
>
> Dear Reviewer YpkJ:
>
> We sincerely thank you for your valuable review and your high recognition of our motivation, writing quality, and experimental results. Your positive feedback is deeply encouraging and means a great deal to us.Below, we provide detailed responses to your insightful and constructive comments.We have revised the corresponding content in the main text or added new discussions in the appendix. When modified in the main text, the changes are marked in blue; when an entire new section is added in the appendix, the title is changed to blue.
>
> Throughout the discussion phase, we genuinely look forward to your feedback and are fully committed to addressing any remaining concerns. If our responses have properly resolved your questions, we would be deeply grateful if you would consider raising your score. If anything remains unclear or unconvincing, we would truly appreciate your further guidance, and we stand ready to engage promptly and constructively.
>
> >W1:The paper states that STPO reduces the number of tokens. For very challenging problems such as AIME 2024, the model may need multiple switches to find the correct reasoning path, indicating that exploration is valuable. Does ST risk over-penalizing reasonable exploration and switching, and to what extent might this affect the model’s ability to initially explore diverse reasoning strategies?
>
> Thank you for your valuable question.
>
> **TL;DR:**
>
> ST optimizes exploration efficiency by targeting Invalid Switching (abandoning correct thought) while retaining Valid Switching (abandoning incorrect thought). Experiments show a significantly larger reduction in invalid switches, which is the key to maintaining accuracy and reducing tokens. Furthermore, on highly difficult tasks like AIME 2024, the number of valid switches increased by 69.4%, demonstrating that STPO enhances, rather than impairs, reasonable exploration ability. We further discuss the impact of ST on the model's exploration capability and provide specific experimental data in Appendix C, and reference it in lines 376-377 of Section 4.4.2.
>
> **Detailed Response:**
> First, we would like to clearly define the two types of switching behaviors we focus on:
>
> * **Valid Switching (Reasonable Exploration):** Switching from an incorrect thought ($F$) to the next thought, which represents the model undertaking necessary exploration.
> * **Invalid Switching (Unwarranted Abandonment):** Switching from a correct thought ($T$) to the next thought, which represents the model failing to identify and stick to the correct reasoning path.
>
> The core objective of STPO is to reduce the second type of "invalid switching," thereby improving the persistence and efficiency of reasoning, while retaining and even optimizing the ability for the first type of "valid switching."
>
> ---
>
> We use a simplified example to intuitively demonstrate the effect of STPO. Let $T$ be a correct thought and $F$ be an incorrect thought. The model's response to a correctly answered problem, before and after ST, is shown below.
>
> ### Table: Comparison of Reasoning Chains Before and After ST
> | Stage | Response Chain | Total Switches | Invalid Switches | Valid Switches |
> | :--- | :--- | :---: | :---: | :---: |
> | **Before ST** | $F_1 \to T_1 \to T_2 \to F_2 \to T_3 \to T_4 \to \text{Correct}$ | 5 | 3 | 2 |
> | **After ST** | $F_1 \to T_1 \to \text{Correct}$ | 1 | 0 | 1 |
>
> By penalizing the act of switching out of a correct thought, STPO makes the model learn to stick to the correct reasoning path, thereby reducing the inefficient 5 switches to an efficient 1 switch. Crucially, it still retains the model's exploratory ability to switch from an incorrect thought ($F$) to a correct thought ($T$). The reduction in the number of valid switches is not due to impaired exploration ability, but because the model has already persisted in the initial correct thought and outputted the answer. This demonstrates that the benefit of STPO stems entirely from reducing ineffective switching, without compromising reasonable exploration ability.
>
> ---
>
> Next, we provide specific experimental data to prove this point:
>
> Comparison of Number of Total Switches (NTS), Invalid Switches (NIS), and Valid Switches (NVS) generated by the model before and after applying ST.
>
> | Model | Status | NTS (MATH500) | NIS (MATH500) | NVS (MATH500) | NTS (AIME 2024) | NIS (AIME 2024) | NVS (AIME 2024) |
> | :--- | :--- | :---: | :---: | :---: | :---: | :---: | :---: |
> | **DeepSeek-R1-Distill-Qwen-1.5B** | Vanilla | 3882 | 2130 | 1752 | 2849 | 412 | 2377 |
> | | Steady Thought | 2161&nbsp;(-44.3%) | 872&nbsp;(-59.1%) | 1289&nbsp;(-26.4%) | 4372&nbsp;(+53.5%) | 346&nbsp;(-16.0%) | 4026&nbsp;(+69.4%) |
> | **Qwen3-8B** | Vanilla | 1928 | 1408 | 520 | 3934 | 1779 | 2155 |
> | | Steady Thought | 747&nbsp;(-61.3%) | 505&nbsp;(-64.1%) | 242&nbsp;(-53.5%) | 2189&nbsp;(-44.4%) | 854&nbsp;(-52.0%) | 1355&nbsp;(-37.1%) |
>
> (to be continued...)

---

> ### Author Response · Authors · 2025-11-24
> **Response to Reviewer YpkJ---Part 2/4**
>
> Across all experiments, the magnitude of the decrease in invalid switches is **significantly larger** than the decrease in valid switches. This proves that the maintenance of accuracy and the reduction in token count primarily result from the suppression of unwarranted abandonment.
>
> Another important piece of evidence that STPO does not impair the ability to explore diverse reasoning strategies is that, when the weaker **1.5B model** addresses the highly difficult **AIME24** dataset, the number of effective switches **increases by 69.4\%**. This indicates that when the model encounters extremely challenging problems, STPO does not impede its reasonable exploration.
>
> ---
>
> In summary, STPO does not over-penalize exploration; instead, it optimizes exploration efficiency by rewarding persistence in the correct thought.

---

> ### Author Response · Authors · 2025-11-24
> **Response to Reviewer YpkJ---Part 3/4**
>
> >W2:The core preprocessing step, thought segmentation, relies on entropy-based detection and predefined thresholds. Although the authors mention hyperparameter tuning, there is a lack of analysis on the stability and robustness of these thresholds across different model scales (1.5B vs. 8B) and task types (mathematics vs. programming).
>
> Thank you for your valuable comments.
>
> **TL;DR:**
>
> Thresholds are Model-Specific but Task-Robust. Due to architectural differences, we use distinct optimal entropy thresholds for the 1.5B (3.0) and 8B (1.5) models. Experiments confirm that this model-specific optimal threshold is highly robust across diverse task types (Math and Code), consistently maximizing accuracy and minimizing token usage, which ensures high-quality thought segmentation data.We discuss threshold settings on more models in Appendix D and reference it in Section 4.4.3.
>
> **Detailed Response:**
>
> First, we want to clarify that due to the significant differences in scale and architecture among LLMs, it is reasonable to set an entropy threshold tailored to each model's characteristics. Therefore, we set different entropy thresholds for the 1.5B model and the 8B model, and this threshold proves to be robust across different task types for the same model.
> We now provide specific experimental data to demonstrate this point:
>
> ### Comparison of thought segmentation results under varying thresholds.
>  **NT**: number of segmented thoughts.
>
> | Base model | Threshold | NT | MATH500 (Acc(\%)$\uparrow$) | MATH500 (Tokens$\downarrow$) | AIME 2024 (Acc(\%)$\uparrow$) | AIME 2024 (Tokens$\downarrow$) | LiveCode (Acc(\%)$\uparrow$) | LiveCode (Tokens$\downarrow$) |
> | :--- | :---: | :---: | :---: | :---: | :---: | :---: | :---: | :---: |
> | DeepSeek-R1-Distill-Qwen-1.5B | 2.8 | 20650 | 83.4 | 3854 | 29.2 | 9252 | **32.9** | 7427 |
> | | **3.0** | 10444 | **84.4** | **2809** | **31.2** | **8606** | 32.4 | **7398** |
> | | 3.2 | 4355 | 83.6 | 3657 | 28.3 | 10516 | 32.7 | 8913 |
> | Qwen3-8B | 1.3 | 18792 | 93.8 | **2392** | 65.0 | 9048 | 76.0 | 6055 |
> | | **1.5** | 11528 | **94.4** | 2869 | **65.8** | **8742** | **77.1** | **5759** |
> | | 1.7 | 5706 | 94.2 | 3081 | 64.2 | 9331 | 74.3 | 5953 |
>
> The experimental results indicate that when the threshold is reasonably set, high-quality data can be generated, enabling the model to effectively learn the thought-level optimization approach.

---

> ### Author Response · Authors · 2025-11-24
> **Response to Reviewer YpkJ---Part 4/4**
>
> >W3:In the thought completion stage, the model prevents switching by directly lowering the logits of trigger words such as “wait” and “alternatively.” This heuristic intervention might contradict the goal of STPO, which aims to implicitly suppress such words through learning.
>
> Thank you for your insightful question.
>
> **TL;DR:**
>
> The roles of Logits intervention and the STPO learning mechanism are separate and non-contradictory. Logits Intervention is used only during the data generation phase to enforce persistence and synthesize high-quality Chosen training samples. The STPO learning mechanism then uses these samples in the model optimization phase to train the model to internalize the suppression of unnecessary switching. Post-training, the model requires no external Logits intervention during inference.
>
> **Detailed Response:**
>
> Logits intervention is not a substitute for the STPO learning mechanism, but rather an indispensable pre-processing step for generating high-quality training data.
>
> Specifically:
>
> During the Thought Completion Stage, we enforce the model to persist in the current reasoning path by directly reducing the Logits of switching trigger words such as "wait" or "alternatively." The sole purpose of this operation is to synthesize high-quality training chosen samples that exhibit "persistence in correct reasoning." These samples represent the ideal behavior the model should adopt—continuously deepening the current promising thought, instead of switching excessively.
>
> The STPO learns to distinguish between "sustained and successful thoughts" (chosen samples) and "excessively switched thoughts" (rejected samples). The model will learn efficient and rational reasoning patterns, thereby internalizing the inhibition of unnecessary thought switching.
> Once training is complete, the model no longer requires any external Logits intervention during the inference stage, allowing it to autonomously achieve persistence in high-potential thoughts.
>
> In summary, there is a clear division of roles between Logits intervention and the STPO learning mechanism:
>
> **Logits Intervention** is a data synthesis method operating during the training data generation phase.
>
> **The STPO** is a learning and internalization mechanism operating during the model optimization phase.

---

### Meta-Review · Area_Chair_8UZG · 2026-01-06

**Summary:**

In the original reviews, the reviewers raised several key concerns about the paper: Ypkj questioned whether the method might over-penalize reasonable exploration, the robustness of entropy-based segmentation thresholds, and potential conflicts between heuristic logits intervention and the learning objective. nhq2 doubted the justification for using entropy in segmentation, the technical depth and novelty of the method, and found the results in Table 2 confusing. QcuC highlighted limitations in generalization due to reliance on explicit switch tokens, robustness of thresholds, computational overhead, the assumption that a single early thought can solve complex problems, and limited experimental scope. Knbc strongly critiqued the lack of novelty, absence of qualitative evidence for reasoning stabilization, and insufficient mechanistic proof that improvements stem from thought-level optimization rather than shorter decoding or regularization.

I also feel that the novelty of the proposed method is limited as the technical components are not innovative and no new research insights are brought by the paper.

**Reviewer Concerns:**

The authors' rebuttal effectively addressed many concerns: they provided new experiments showing increased valid switching, demonstrated threshold robustness across tasks, clarified that logits intervention is only for data synthesis and not in conflict with STPO, added a 14B model and GSM8K results to broaden scope, and included qualitative examples of stabilized reasoning chains. However, some concerns remain less fully resolved: the novelty claim still relies heavily on reframing and integrating existing techniques rather than introducing fundamentally new algorithms, and while mechanistic evidence was strengthened, some reviewers may still view the approach as an incremental combination of prior methods rather than a groundbreaking framework.

**Reviewer Scores:**

Reviewer Ypkj (score: 6): Likely would have maintained or slightly raised the score, given the clear experimental responses showing ST enhances valid exploration and the clarification on logits intervention.

Reviewer nhq2 (score: 4): Probably would have raised the score to a weak accept due to the added justification for entropy and clearer explanation of Table 2.

Reviewer QcuC (score: 4): Likely would have increased the score to a weak accept, as the rebuttal expanded the experimental scope and addressed computational cost and generalization concerns with new data and reasoning.

Reviewer Knbc (score: 2): Likely would have maintained the Reject score due to the novelty issue.

---

### Decision · Program_Chairs · 2026-01-26

Reject